# Rapid reappearance of air pollution after cold air outbreaks in northern and eastern China

Qian Liu[1], Guixing Chen[1], Lifang Sheng[2], Toshiki Iwasaki[3]

[1]School of Atmospheric Sciences, Sun Yat-sen University, and Southern Marine Science and Engineering Guangdong Laboratory (Zhuhai), Zhuhai, 519082, China
[2]Department of Marine Meteorology, College of Oceanic and Atmospheric Sciences, Ocean University of China, Qingdao, 266100, China
[3]Department of Geophysics, Graduate School of Science, Tohoku University, Sendai, 980-8577, Japan

*Correspondence to*: Guixing Chen (chenguixing@mail.sysu.edu.cn)

**Abstract.** The cold air outbreak (CAO) is the most important way to reduce air pollution during the winter over northern and eastern China. However, a rapid reappearance of air pollution is usually observed during its decay phase. Is there any relationship between the reappearance of air pollution and the properties of CAO? To address this issue, we investigated the possible connection between air pollution reappearance and CAO by quantifying the properties of the residual cold airmass after CAO. Based on the analyses in recent winters (2014–2022), we found that the rapid reappearance of air pollution in the CAO decay phase has an occurrence frequency of 63%, and the air quality in more than 50% of CAOs worsens than that before CAO. The reappearance of air pollution tends to occur in the residual cold airmass with weak horizontal flux during the first two days after CAO. By categorizing the CAOs into groups of rapid and slow air pollution reappearance, we found that the residual cold airmass with moderate depth of 150–180 hPa, large negative heat content and small slopes of isentropes is favorable for the rapid reappearance of air pollution. Among these factors, the cold airmass depth is highly consistent with the mixing layer height, below which contains most air pollutants; the negative heat content and slope of isentropes in cold airmass jointly determine the intensity of low-level vertical stability. The rapid reappearance of air pollution is also attributed to the maintenance of residual cold airmass and the above conditions, which is mainly regulated by the dynamic transport process rather than diabatic cooling or heating. Furthermore, analysis of the large-scale circulation of CAOs in their initial stage shows that the anticyclonic (cyclonic) pattern in northern Siberia (northeastern Asia) can be recognized as a precursor for the rapid (slow) reappearance of air pollution after the CAO.

# 1 Introduction

Air pollution is one of the main meteorological disasters in China during the boreal winter, threatening public health, the environment and economic activities (Guan et al., 2016; Xu et al., 2013; Tie et al., 2016). Northern and eastern China (NEC), experiences the most pollution, which mainly is haze pollution, resulting in attention from researchers. Statistics show that air pollution or haze events in NEC have an occurrence frequency of approximately 20–40 days during winters since 2010 (Ding and Liu, 2014; Chen and Wang, 2015; Yang et al., 2016; Dang and Liao, 2019; Zhang et al., 2020). In some cities, such as Beijing, the occurrence frequency approaches 60 days (Li et al., 2021). Moreover, severe air pollution has also occurred frequently over NEC in recent years, during which the air quality index usually exceeds the upper limit. In some widely reported air pollution events during January 2013, December 2015 and December 2016, the regional daily mean $PM_{2.5}$ concentration exceeds 500 µg m$^{-3}$ (Zhang et al., 2014; Sun et al., 2016; Yin and Wang, 2017). Even when local emission is obviously reduced during COVID-19 lockdown, severe air pollution still occurs in North China (Zhao et al., 2020). Faced with the demands of improving air quality, progress in understanding the variations in air pollution is of great importance.

The formation of air pollution in NEC can be attributed to heavy emissions, topography and unfavorable meteorological conditions (Zhang et al., 2013; Cao et al., 2015; Yang et al., 2016; Dang and Liao, 2019; An et al., 2019). The recent COVID-19 lockdowns also provide a unique opportunity for studying the complex chemical effects of air pollution as well as meteorology (Le et al., 2020; Wang et al., 2020). As emissions and topography do not vary much from day to day, meteorological conditions play an important role in impacting variations in air pollution on the synoptic time-scale (Kang et al., 2019; Ma et al., 2021; Zhang et al., 2021). Previous studies showed that weak surface winds, high relative humidity, enhanced temperature inversion, low atmospheric boundary layer height, and less precipitation facilitate the accumulation and maintenance of air pollutants (Ding and Liu, 2014; Wang and Chen, 2016; Li et al., 2017). These unfavorable meteorological conditions usually occur in stagnant weather controlled by high-pressure systems or anomalous anticyclonic patterns (Yang et al., 2018; Zhong et al., 2019; An et al., 2020).

Cold air outbreaks (CAOs) are generally considered to be conducive to the removal of air pollution in NEC (Qu et al., 2015; Li et al., 2016; Liu et al., 2017; Hu et al., 2018; Zhang et al., 2021). However, our recent study shows that air pollutant concentrations sometimes experienced a rapid increase after CAO, forming a new round of air pollution with even higher pollutant concentrations than the period before CAO (Liu et al., 2019). During the reappearance of air pollution, the regional mean pollutant concentrations increased as much as 2.8 times than concentrations reduced by CAO. Some studies also provide clues for the reappearance of air pollution after CAO. Under high emissions conditions, the removal of air pollution caused by strong surface winds is usually maintained for only approximately 1 day and then returns to the polluted state rapidly (Liu et al., 2016). Wang et al. (2014) shows that during persistent heavy air pollution in January 2013, cold air activity could only temporarily reduce the air pollutant concentration. The reappearance of air pollution is usually characterized by explosive growth, with air pollutant concentrations increasing from a few to tens and even hundreds of micrograms per cubic meter within 1 day (Sun et al., 2016). These results suggest that a considerable portion of air pollution in NEC can be linked to the

decay of the CAO. Many previous studies have noted the reappearance of air pollution after CAO, while few studies have made statistics on the reappeared air pollution. So far, the quantitative relationship and relevant physical mechanisms between air pollution reappearance and CAO properties are still unclear. To explore the mechanisms modulating rapid increases in air pollutants, further analysis showing the detailed features and structures of CAOs, especially during their decay stages, is required. Additionally, since CAOs are easy to identify and predict in their initial stages, they also have potential as precursory

signals for the burst of severe air pollution.

In this study, a quantitative measurement of cold airmass (Iwasaki et al., 2014) is adopted to identify CAO and its dynamic/thermodynamic characteristics. Based on the analyses of CAOs and accompanying variations in air pollution during recent winters, our study will verify the generic existence of air pollution reappearance after CAO. Then, we will clarify the CAO with what characteristics, such as depth, horizontal flux, coldness and large-scale circulation pattern of cold airmass,

may result in the rapid reappearance of air pollution. The rest of this study is organized as follow. The data and methods used in this study are described in Section 2. Statistical features of air pollution reappearance, as well as its association with CAO evolution, are examined in Section 3. The key features of CAO affecting the air pollution reappearance and the involved physical mechanisms are discussed in Section 4. Section 5 further investigates the large-scale circulation pattern of the CAO. A short summary is presented in Section 6.

## 2 Data and methods

### 2.1 Data

To identify the CAO events, we used isobaric analysis data of Japanese 55-year reanalysis (JRA-55). This dataset is freely available at JMA (http://search.diasjp.net/en/dataset/JRA55) and NCAR (https://rda.ucar.edu/datasets/ds628.0/). The JRA-55 has a horizontal resolution of 1.25° with 37 vertical pressure levels and a time interval of 6 hours (00, 06, 12 and 18 UTC).

Previous studies have shown that JRA-55 has good performance in identifying and describing CAOs over East Asia (Yamaguchi et al., 2019; Liu et al., 2020; Liu et al., 2021).

The variation in air pollutant concentration in this study is described using the air quality index (AQI) from the Ministry of Ecology and Environment of the People Republic of China. The AQI quantifies overall air quality based on the primary air pollutant in the monitored area, with observations of 1-hour frequency. Data from 475 stations (in 98 cities) across NEC (30°–

40°N, 114°–122°E) covering the winters (November to March) from 2014/2015 to 2021/2022 were used. The determination of study area (NEC), including parts of both northern China and eastern China, is based on the characteristics of both air pollution and cold air activity. The air pollution (AQI > 100) mainly occurs between the latitudes of 30°N and 40°N, and the CAOs also usually affect areas around 30°N (Figure omitted). The AQI stations in the study area have a relative even distribution (figure omitted). The stations are located in not only urban areas but also suburban and rural areas. It should be

noted that the daily mean values of AQI and another well-known air pollution index of $PM_{2.5}$ concentration (primary air

pollutant of haze) has a high correlation coefficient of 0.96 in study period. The key results of this study are in a good agreement with results based on PM$_{2.5}$ concentration (figure omitted).

The sounding data obtained from the University of Wyoming provides a direct detection of atmospheric vertical profile, which would help to explain the changes of AQI observation. Four sounding stations were selected in NEC: Beijing (39.8°N, 116.5°E), Zhangqiu (36.7°N, 117.6°E), Nanjing (31.9°N, 118.9°E) and Baoshan (31.4°N, 121.5°E). Observation times were 00 and 12 UTC (08 and 20 LT). The vertical resolution of the sounding data is comparable to that of JRA-55 model grid reanalysis data. For example, sounding data at Beijing station has 60–70 levels in total and 8–14 levels below 850 hPa during a CAO event of 14–17 Dec 2016. The data sources of all variables used in this study are listed in Table 1.

**Table 1:** Data source of variables used in this study. Here, $u$: zonal winds, $v$: meridional winds, $T$: air temperature, $\Phi$: geopotential height, $p_s$: surface pressure.

| Data source | Variables |
|---|---|
| JRA-55 reanalysis data (Japan Meteorological Agency) | $u, v, T, \Phi$ (1000~100 hPa) and $p_s$ |
| Air quality monitoring data (Ministry of Ecology and Environment of the People Republic of China) | AQI |
| Radio sounding data (University of Wyoming) | $u, v, T, \Phi$ (surface to 100 hPa) |

**2.2 Isentropic analysis of cold air outbreak**

The identification and quantification of cold airmass and its outbreaks are performed by an isentropic analysis method. The cold airmass is defined as a layer of airmass below the threshold isentropic surface of $\theta_T = 280$ K, which includes most of the equatorward cold airmass flow (Iwasaki et al., 2014). Thus, the depth of the cold airmass ($DP$, unit: hPa) can be described as the pressure difference between the ground surface ($p_s$) and the $\theta_T$ surface ($p(\theta_T)$),

$$DP = p_s - p(\theta_T). \tag{1}$$

The horizontal flux of the cold airmass ($F$, unit: hPa m s$^{-1}$) is given by the vertical integration of the horizontal wind ($v$) and the pressure ($p$),

$$F = \int_{p(\theta_T)}^{p_s} v \, dp. \tag{2}$$

The negative heat content ($NHC$, unit: K hPa), reflecting the coldness of the cold airmass, is defined as a vertical integration of the potential temperature difference ($\theta_T - \theta$) and pressure ($p$),

$$NHC = \int_{p(\theta_T)}^{p_s} (\theta_T - \theta) \, dp. \tag{3}$$

Based on the above method, the CAO can be identified by the regional mean cold airmass depth. Previous studies showed that the cold airmass depth usually experiences a sharp increase and reaches a relatively high value during the outbreak (Shoji et

al., 2014; Liu et al., 2019; Liu et al., 2021). Thus, the CAO in this study is identified when the regional mean cold airmass depth exceeds 166.8 hPa, which is the sum of mean value (77.3 hPa) and standard deviation (89.5 hPa) of cold airmass depth on all winter days. According to the above criteria, 52 CAOs are identified over the 8 winters.

The CAO is further divided into 3 periods. Figure 1 shows the temporal evolution of the cold airmass depth in NEC during a typical CAO in March 2016. The onset of a CAO event, which is the first day when regional mean cold airmass depth exceeds

the threshold (166.8 hPa), is described as the day 0. The period during the CAO starts from the onset day (day 0) and ends at the day when cold airmass depth falling below the threshold (day +2 in CAO event plotted in Figure 1). The period before CAO is defined as the two days before onset day to the onset day (days −2 to 0). The period after CAO, which is also called the decay phase, is defined as the three days after cold airmass depth falling below the threshold (days +2 to +4 in CAO event plotted in Figure 1). The period after CAO varies with events, since the end date of period during the CAO is different in the

selected events. It should be noted that the cold airmass in the period after CAO is described as the residual cold airmass in this study.

## 2.3 Measurement of air pollution reappearance and mixing layer height

The description of air pollution reappearance is based on the AQI in different periods of the CAO, as shown in Figure 1. In the period before CAO, the AQI decreases from a polluted state to a relatively clean state. The AQI usually reaches its minimum

in the period during CAO. Then, the AQI increases and results in a new round of air pollution in the period after CAO. Thus, we use the maximum AQI in the period before CAO ($AQI\_b$) and the maximum AQI in the period after CAO ($AQI\_a$) to represent the air pollution before CAO and the reappeared air pollution, respectively. The minimum AQI in the period during CAO ($AQI\_d$) is used to represent the clean state between the air pollutions before and after the CAO.

Based on the above analysis, the increase of AQI ($IA$) is defined as follows to describe the deterioration rate of air pollution

in its reappearance:

$$IA = AQI\_a - AQI\_d. \tag{4}$$

A rapid reappearance of air pollution is supposed to have an $IA$ larger than 34.3, which is the standard deviation of the AQI daily variation during the 5 winters. A reappearance index of air pollution ($RI$) is also defined to describe the relative intensity of reappeared air pollution compared to the air pollution before CAO,

$$RI = AQI\_a/AQI\_b, \tag{5}$$

where $RI$ larger (smaller) than 1 denotes that the AQI after CAO is worse (better) than that before CAO.

The mixing layer height is determined from the analysis of the profile of the bulk Richardson number ($R_B$) using sounding data. The calculation method for $R_B$ is derived from Tang et al. (2016) as follows:

$$R_B = \frac{g\,\Delta\theta\,\Delta Z}{\theta\,((\Delta u)^2 + (\Delta v)^2)}, \tag{6}$$

where $g$, $\theta$, $Z$, $u$ and $v$ indicate the gravity acceleration, potential temperature, height, zonal wind and meridional wind, respectively. According to the profiles of $R_B$ and $\theta$, the observations can be separated into a convective state and a stable state. In this study, we found that all the sounding observations are in the stable state with no obvious low-level jet. Therefore, the mixing layer height (MLH) is defined as the altitude at which $R_B$ is greater than 1.

## 3 Characteristics of air pollution reappearance after CAO

### 3.1 Statistics of AQI after CAO

To ensure the existence of the rapid reappearance of air pollution after CAO, the relationship of daily changes in AQI and cold airmass depth is divided into four quadrants in Figure 2. In winter, 62.1% of the decrease in the AQI occurs when the cold airmass depth increases, which corresponds to the period before and during the CAO. On the other hand, most of the AQI increase (68.6%) was accompanied by a cold airmass decrease, which usually occurred in the period after the CAO. Based on

these results, the AQI variation changes from decreasing to increasing throughout the CAO, suggesting the existence of air pollution reappearance after the CAO. We also noticed that most of the heavy air pollution was in the quadrant of cold airmass depth decrease, suggesting that the retreat and dissipation of cold airmass may be an important meteorological factor for the formation of heavy air pollution.

Based on the 52 CAOs selected in this study, the statistical characteristics of the air pollution reappearance after CAO can be

demonstrated. Figure 3a compares the AQI daily change rate in the period after CAO to that on all winter days. During the period after CAO, the increase in the AQI has an occurrence frequency of approximately 80%, which is much higher than that on all winter days (56%). In particular, the relatively high increase rate ($> 20$ day$^{-1}$) in the period after CAO has a frequency of approximately twice as high as that on all winter days. Such a frequent rapid increase in the AQI after CAO could induce the reappearance of air pollution and even may result in a worse AQI compared to the AQI before CAO. Using the Eq. (5),

Figure 3b shows the reappearance rate of air pollution ($RI$) after CAO, whose value greater (smaller) than 1 denotes that the AQI after CAO is worse (better) than the AQI before CAO. Based on the definitions of periods before and after CAO in our study, more than 50% of the CAOs are found to show a worse AQI after reappearance. In some extreme events, the AQI after CAO could be twice as high as the AQI before CAO. These facts show that even if the CAO brings several days of clean air, it may lead to a deterioration of the AQI to the polluted or an even worse state.

Figure 3c shows that the $IA$ (position of arrows on x-axis) in all of CAO events have a positive value, which means in AQI has experienced an increase after CAO. This result confirms the reappearance of air pollution is a common phenomenon. Considering the different characteristics of AQI variations after the CAO, the CAOs can be categorized into a "slow reappearance" group and a "rapid reappearance" group according to $IA$ defined by Eq. (4). Among the 52 CAOs, 19 (36.5%) show a slow reappearance of air pollution after CAO (CAO_slow), with an increase in the AQI of less than 1 standard deviation

of the AQI daily variation ($IA \leq 34.3$). Note that not all CAOs are accompanied by a rapid air pollution reappearance. The remaining 33 CAOs (63.5%) featured a rapid reappearance of air pollution (CAO_rapid). In Figure 3c, the arrow on y-axis has

a similar meaning with *RI*, indicating whether the air quality after CAO will get better or worse than that before CAO. The position of arrowhead and tail on y-axis denote the $AQI\_b$ and $AQI\_a$, respectively. The orange up arrow (blue down arrow) represents the reappeared air pollution has a heavier (lighter) degree than the air pollution before CAO. Result shows that most CAO_slow events have a better AQI than that before CAO, while nearly two-thirds of CAO_rapid events face a worse AQI. The above differences in AQI changes after CAO suggest that the air pollution reappears rapidly or slowly may be modulated by the detailed characteristics of CAO.

## 3.2 Relationship between the reappearance of air pollution and CAO

In terms of spatial averages over NEC, Figure 4 shows large differences in the cold airmass depth and AQI between CAO_rapid and CAO_slow events. For CAO_rapid events (Figure 4a), the cold airmass depth increases from day −2 and reaches its maximum on day 0. Then, the cold airmass depth gradually decreases to approximately 100 hPa in the following 3 days. During the whole life cycle of the CAO_rapid event, the AQI shows a reversed variation to cold airmass depth. From day −2 to day 0, the AQI fell to a moderate level below 70. However, shortly after, the AQI increased rapidly to nearly 130 on day +2, which was worse than the AQI before the CAO. For CAO_slow events (Figure 4b), the cold airmass depth also increases from day −2 to day 0 and reaches a value similar to that in CAO_rapid events. After day 0, the cold airmass depth drops to the minimum on day +2 at a much faster rate than that in CAO_rapid events and rises again from day +3. During the CAO_slow events, the AQI still varies conversely to the cold airmass depth. However, the decrease in the AQI starts on day 0, which is later than that in CAO_rapid events, and the variation in the AQI also has a relatively small amplitude. Such differences in the removal of air pollution may be associated with the thermal/dynamical features of CAO (Liu et al., 2019). The contrasts of the CAO_rapid and CAO_slow events suggest that there may be two favorable conditions for the rapid air pollution reappearance: one is the relatively slow decrease of cold airmass depth, the other is the longer time period before the next CAO.

We further investigate the spatial-temporal collocation of the AQI and cold airmass distributions during and after the CAO. For CAO_rapid events, a thick layer of cold airmass accompanying the strong southeastward flux is observed in NEC on day 0 (Figures 5a–5b). At this stage, the AQI decreased markedly, especially in the eastern part of NEC. From day +1 to day +3, the southeastward flux of cold airmass weakens, and a quasi-stationary residual cold airmass occupies NEC (Figures 5c–5h). During this period, the AQI increased rapidly and even exceeded 150 (middle level air pollution) at a number of stations. The worst AQI usually occurs in the region with a cold airmass depth of 50–150 hPa. For CAO_slow events on day 0, the cold airmass depth has a distribution similar to that in CAO_rapid events (Figures 6a–6b). However, the cold airmass fluxes in CAO_slow events contain a larger eastward component than those in CAO_rapid events. Such an enhanced eastward flux leads to the quick eastward retreat of residual cold airmass on days +1 and +2 (Figures 6c–6f). During this period, an increase in the AQI is also observed in the area covered by the residual cold airmass. On day +3, the cold airmass enhances again in the northern part of NEC and thus ends the reappearance of air pollution (Figures 6g–6h). According to the above results, we found that the reappearance of air pollution tends to occur in the residual cold airmass with nearly no horizontal flux during the decay period of the CAO.

## 4 Factors affecting the rapid reappearance of air pollution

To determine the dominant feature of CAO controlling the reappearance of air pollution, Table 2 compares the dynamic and thermal-dynamic properties of cold airmass in CAO_rapid and CAO_slow events. The cold airmass properties are averaged in the period from day +1 to +2, which corresponds to the rapid increase period of the AQI in Figure 4. The results reveal that the magnitude of cold airmass flux shows a relatively small difference between the two types of CAO. However, large differences exist in the depth and NHC of cold airmass, which may be the key factors affecting the air pollution reappearance. The cold airmass in CAO_rapid events (143.4 hPa and 767.1 K hPa) is thicker and colder than that in CAO_slow events (111.9 hPa and 467.0 K hPa). To explore how the depth and NHC of cold airmass modulate the air pollution reappearance, a CAO_rapid event (14–17 December 2016) and a CAO_slow event (11–14 February 2018) are selected for further analysis.

**Table 2:** Averaged AQI change rate and cold airmass properties during the period from day +1 to +2.

|  | Change rate of AQI (day$^{-1}$) | Cold airmass depth (hPa) | Magnitude of cold airmass flux (hPa m s$^{-1}$) | Negative heat content (K hPa) |
|---|---|---|---|---|
| CAO_rapid | 22.8 | 143.4 | 909.9 | 767.1 |
| CAO_slow | 9.3 | 111.9 | 763.7 | 467.0 |

### 4.1 Cold airmass depth

Figures 7 and 8 examine the profile of potential temperatures below 3 km at the four stations (see Section 2.1) under the effect of the CAO. During the CAO_rapid event on day 0, the lapse rate of potential temperature is small from the near ground level to the height of approximately 1.5 km (Figure 7a). Above the height of 1.5 km, the lapse rate suddenly increases, forming an air layer with a large vertical gradient of potential temperatures. Such a large vertical gradient of potential temperature denotes a strong stable stratification, weakening the vertical mixing of air pollutants to the upper level. In the case of high emissions in NEC (Zhang et al., 2012; Gao et al., 2016), atmospheric conditions such as this are favorable for the accumulation of air pollutants in the near surface atmosphere (Zhang et al., 2022). On the other hand, the large gradient layer can be recognized as the upper boundary of the cold airmass since the cold airmass features strong thermal contrast to its surroundings. This upper boundary of cold airmass acts as a lid confining the air pollutant in the residual cold airmass. Previous studies also show that air pollutants usually increase and are trapped in the residual cold airmass during the period after CAO (Kang et al., 2019; Liu et al. 2019). From day 0 to +1, the height of the large gradient layer is maintained at approximately 1.5 km, and then it slowly drops to 0.5 km in the following 2 days. Such a temporal evolution is highly consistent with the evolution of cold airmass depth (Figures 7a and 7b). In the same period, the AQI increases nearly 2 times from 50 to 150.

The low-level atmosphere with a small lapse rate of potential temperature in Figure 7a may represent the mixing layer (or boundary layer), featuring convection, vertical mixing and dry-adiabatic lapse rates. The mixing layer could greatly impact the accumulation and dissipation of air pollutants (Holzworth, 1967; Tang et al., 2016). Many studies have reported that the MLH is a good indicator of air quality (Su et al., 2018; Gui et al., 2019). However, the calculation of MLH is somewhat complex (Eq. (6)), and the MLH calculated by most mainstream reanalysis datasets has obvious bias compared to sounding data (Guo et al., 2021). Figure 7b shows the MLH at the four stations at 12 UTC. The MLH and cold airmass depth have good agreement in both values and temporal variations. This is because the vertical structures of temperature and wind in the low-level atmosphere are mainly forced by the cold airmass during CAO. Thus, the residual cold airmass happens to be the mixing layer and modulates the evolution of the AQI. In addition, the calculation of cold airmass depth using reanalysis data is much easier than the calculation of MLH using sounding or other observation data, suggesting that the cold airmass depth may be a good substitute for MLH in periods during and after the CAO.

In the CAO_slow event on day 0, the potential temperature has a profile similar to that in the CAO_rapid event (Figure 8a). The large gradient layer is located at approximately 2.5 km, which corresponds to the cold airmass depth and MLH (Figure 8b). Such features in cold airmass and atmospheric stratification continue to 00 UTC on day +1. During this period, the AQI shows no increase due to the high MLH. From day +1 to +2, the cold airmass depth decreases rapidly and almost disappears in the southern part of NEC, leading to the disappearance of the large gradient layer in low troposphere. It should be noted that the mixing layer at this time is not forced by residual cold airmass since it is absence. As the MLH decreases to 1.5 km, the AQI experiences a relatively slow increase. On day +3, the cold airmass depth increases again, causing an increase in the MLH and a decrease in the AQI.

Based on the analyses of a CAO_rapid event and a CAO_slow event, we conclude that the intrusion of cold airmass forces the lower troposphere into a stable stratification that is favorable for the increase in the AQI. Thus, the depth and persistence of the residual cold airmass were thought to be two factors that influence the deterioration rate of air pollution. It is clear that a longer persistence of residual cold airmass is favorable for the rapid reappearance of air pollution, while the relation between cold airmass depth and the AQI increase rate is quite complex. Regarding the rapid reappearance of air pollution, the cold airmass depth is not the lower the better. In CAO_rapid events, for example (Figure 5), the increase in the AQI is less obvious in the region to the south of NEC (25°–30°N) with a cold airmass depth thinner than 100 hPa. Moreover, the AQI increase rate will not vary monotonically with cold airmass depth. The AQIs in Yangtze River Delta (30°–32°N, 118°–122°E) and Bohai Rim (37°–40°N, 117°–122°E) have similar increase rates, although the Bohai Rim has a cold airmass depth much thicker than that of the Yangtze River Delta. Similar results can also be found in CAO_slow events (Figure 6). According to these analyses, Figure 9 presents the relation between cold airmass depth and the AQI change rate. With increasing cold airmass depth, the change rate of the AQI first increases and then slightly decreases. A cold airmass depth that is too thick dilutes the surface air pollutant concentration, but a cold airmass depth that is too thin damages the persistence of the cold airmass. The cold airmass depth in a moderate range of 150–180 hPa (approximately 1.5–1.8 km) is most favorable for the air pollution reappearance,

with a mean increase rate of 26.1 day$^{-1}$. In some extreme cases exceeding the 90$^{th}$ percentile, the AQI increase rate can reach 72.5 day$^{-1}$.

## 4.2 NHC and vertical thermal structure

As another key factor revealed in Table 2, NHC also contributes to air pollution reappearance. According to the definition in Section 2.2 (Eq. (3)), NHC is proportional to the production of cold airmass depth ($DP$) and vertical averaged potential temperature difference ($\Delta\theta = \overline{(\theta_T - \theta)}$). If the potential temperature increases linearly with height, $\Delta\theta$ can be further simplified as ($\theta_T - \theta_s$), where $\theta_s$ is the surface potential temperature. Here, diving the NHC by ($DP$)$^2$ lead us to the following:

$$\frac{NHC}{(DP)^2} \propto \frac{\Delta\theta}{\Delta p}\bigg|_{Cold\ airmass}, \tag{7}$$

where $\Delta p$ is equal to $DP$ when referring to the cold airmass, and $\frac{\Delta\theta}{\Delta p}$ indicates the averaged vertical stability in the cold airmass.

Thus, at a given cold airmass depth ($DP$), a larger NHC will likely result in a more stable stratification, which is favorable for air pollution reappearance. This explains why the NHC is much larger during the decay period in CAO_rapid events than in CAO_slow events (Table 2). On the other hand, the persistence of cold airmass will also benefit from the large NHC, which helps to dampen the reduction effect of diabatic heating.

In addition to NHC, the two selected CAO events show that the vertical thermal structure of cold airmass also influences the stability of low-level troposphere (Figure 10). On the onset day of the CAO_rapid event (Figure 10a), the upper boundary of cold airmass is relatively flat, rising from 900 hPa at 30°N to 770 hPa at 40°N. Inside the cold airmass, the isentropes represented by the potential temperature anomaly are essentially parallel to the upper boundary of cold airmass. Such flat distributions indicate that the vertical gradient is the dominant component of the potential temperature gradient. As a result, a
strong stable layer forms at the upper part of cold airmass spanning 900–700 hPa. Note that this large vertical gradient layer can also be seen in Figure 7a. After the outbreak on day +1 (Figure 10b), the upper boundary of cold airmass remained almost unchanged, and the negative anomaly of potential temperature weakened slightly. The stable layer still exists at the upper part of cold airmass, although its intensity is weaker than that on day 0. For the CAO_slow event on day 0 (Figure 10c), the upper boundary of cold airmass steeply increases from 900 hPa at 30°N to higher than 600 hPa at 40°N. The slope of cold airmass
in the CAO_slow event is more than twice that in the CAO_rapid event. The isentropes in the cold airmass are almost perpendicular below 850 hPa. In this scenario, the vertical gradient of potential temperature is small in the CAO_slow event, even if it has a thicker and colder cold airmass than that in the CAO_rapid event. Accordingly, a stable layer with smaller intensity but higher altitude (800–600 hPa) is observed over the central and southern parts of cold airmass. On day +1 (Figure 10d), the cold airmass depth decreases markedly; however, the slope of isentropes remains large. As a result, the relatively
weak stable layer lowers to approximately 850 hPa.

The differences in vertical atmospheric stability may lead to different increase rates of AQI during air pollution reappearance. Figure 11 compares the evolutions of vertical stability in NEC during the two CAO events. From 00 UTC day 0 to 12 UTC

day +1, the CAO_rapid event has a larger stability than that in the CAO_slow event at both the upper boundary and the entire cold airmass. In particular, the stability of the entire cold airmass in the CAO_rapid event is twice as high as that in the CAO_slow event. During this period, the AQI in the CAO_rapid event increases drastically from 50 to 120 under strong stable stratification, while the AQI in the CAO_slow event remains in the range of 80–100 under weak stable stratification (Figures 7b and 8b). After 12 UTC day +1, the variation in the AQI becomes steady, while the atmospheric stability starts to increase (Figure 11). This may be attributed to the radiative feedback of the air pollutant (Petäjä et al., 2016; Ding et al., 2016). By comparing the two types of CAO events, we find that the NHC and vertical structure of cold airmass jointly affect the low-level vertical stability, which modulates the air pollution reappearance. A cold airmass with a small slope of isentropes and a large NHC is favorable for the rapid reappearance of air pollution.

## 4.3 Impacts of cold airmass features on air pollution reappearance in numerical simulation

To verify the rolls of above-mentioned cold airmass features on air pollution reappearance inferred from the analyses of two selected CAO events, numerical experiments using WRF-Chem version 4.3 are carried out in this section. The domain of simulations is designed with a horizontal grid spacing of 10 km covering most part of East Asia (figure omitted). The FNL data is used as the initial and lateral boundary conditions to drive the meteorological simulation. The MEIC anthropogenic emission inventories are used in the chemical simulation. The main physical and chemical parameterization schemes include the WSM6 microphysics, the MYJ PBL scheme, the RRTM for longwave and shortwave radiation, RADM2-MADE/SORGAM for gas-phase chemical and aerosol schemes.

Figure 12 shows the spatial averaged cold airmass depth, boundary layer height, vertical stability and surface $PM_{2.5}$ concentration during the CAO_rapid and CAO_slow events. The emissions in both two experiments are set as same values in December 2016 to investigate the impacts of meteorological conditions. In experiment of CAO_rapid event, the air pollutant concentration increases rapidly on days 0 and +1 under the condition of the relatively low boundary layer height and strong vertical stability. Such conditions of atmospheric boundary layer are not conducive to the diffusion of air pollutant and tend to induce rapid reappearance of air pollution (Zhang et al., 2014; Liu et al., 2017). In CAO_slow experiment, however, the $PM_{2.5}$ concentration keeps in a low level due to the relatively high boundary layer height and weak vertical stability. In addition, the temporal evolutions of these variables are highly consistent with the observations shown in Figures 7–8 and 11, suggesting both the rapid and slow reappearances of air pollution can be well captured by numerical model.

To further verify the connection between cold airmass properties and boundary layer diffusion conditions as discussed in Sections 4.1 and 4.2, a control experiment (the CAO_rapid event from 14 to 17 Dec 2016) and additional sensitive simulations are also conducted. In sensitive experiments, temperature disturbances are artificially added in the initial field following Bai et al (2019). In NHC_C (NHC_W) experiment, the NHC of cold airmass is increased (decreased) by adding a cold (warm) bubble centered at a height of 0 km. The cold (warm) bubble has a latitudinal radius of 10 km, longitudinal radius of 5 km and a vertical radius of 2 km, with a minimum potential temperature perturbation of −8 (8) K. The temperature disturbance, in a cold bubble for example, is minimized at the center and increases to 0 K following a cosine function over the horizontal and

vertical radius. To increase (decrease) the cold airmass depth in DP_C (DP_W) experiment, the cold (warm) bubble added in the initial field moves to the height of 2 km. Note that the NHC may also change with cold airmass depth in DP_C and DP_W experiments.

Table 3 shows the simulation results averaged in the study area on day 0, when air pollutant has the rapidest increase rate as shown by Figure 12a. In NHC_C and NHC_W experiments, changes in NHC cannot cause an obvious variation in boundary layer height, but can lead to changes in vertical stability. In DP_C and DP_W experiments, despite the changes of NHC and vertical stability, we find that changes in cold airmass depth will result in obvious changes in boundary layer height. These sensitive experiments confirm the main results of Sections 4.1 and 4.2, that is, the properties of CAO could effectively impact the diffusion conditions in atmospheric boundary layer.

**Table 3:** Averaged cold airmass properties and atmospheric boundary layer conditions on day 0 from control and sensitive experiments. Upward (downward) arrows denote the value of sensitive experiment is greater (less) than the value in control experiment.

| Experiment | Cold airmass depth (hPa) | Atmospheric boundary layer height (m) | Negative heat content ($-$ K hPa) | Vertical stability ($10^{-3}$ K m$^{-1}$) |
|---|---|---|---|---|
| Control | 165.6 | 597.2 | 1898.4 | 5.47 |
| NHC_C | 163.9 | 579.9 | 1994.5 ↑ | 6.79 ↑ |
| NHC_W | 167.6 | 661.0 | 1745.9 ↓ | 4.15 ↓ |
| DP_C | 189.4 ↑ | 727.4 ↑ | 2447.9 | 5.98 |
| DP_W | 118.8 ↓ | 519.9 ↓ | 1238.4 | 4.74 |

**4.4 Possible mechanisms governing key features of cold airmass**

Since the properties of the residual cold airmass control the features of air pollution reappearance. It is necessary to investigate the possible mechanisms modifying cold airmass changes during the decay period of the CAO. According to Iwasaki et al. (2014), the total cold airmass conservation equation can be described as follows:

$$\frac{\partial DP}{\partial t} = -\nabla \cdot \int_{p(\theta_T)}^{p_s} v \, dp + G(\theta_T). \tag{8}$$

The change in cold airmass depth is caused by the horizontal convergence/divergence of the cold airmass flux ($-\nabla \cdot \int_{p(\theta_T)}^{p_s} v \, dp$) and the vertical flux crossing the $\theta_T$ surface ($G(\theta_T)$). The former represents the dynamic process, and the latter is related to diabatic cooling and heating.

Based on Eq. (8), Figure 13 compares the composited evolutions of cold airmass changes induced by dynamic and diabatic processes. During the CAO_rapid events (Figure 13a), the decrease in cold airmass is mainly attributed to the dynamic process,

which shows a decreasing effect since 18 UTC day 0 and reaches a maximum of −75.9 hPa day$^{-1}$ at 18 UTC day +1. The effect of the diabatic process on cold airmass exhibits obvious diurnal variation, cooling/increasing the cold airmass at night (12–24 UTC) and heating/decreasing the cold airmass during the daytime (00–12 UTC). On days 0 and +1, the diabatic heating is weak during the daytime, inducing a slight net increase in cold airmass caused by the diabatic process. From day +2, the heating effect intensifies and reduces the cold airmass jointly with the dynamic process. During the decaying period of cold airmass on days +1 and +2, the average effects of dynamic and diabatic processes on cold airmass are −56.9 and 7.4 hPa day$^{-1}$, respectively.

For CAO_slow events (Figure 13b), the dynamic process still dominates in the cold airmass decrease. The average tendencies of dynamic and diabatic processes on days +1 and +2 are −66.7 and 1.5 hPa day$^{-1}$, respectively. The two types of processes show similar variations but larger amplitudes compared to those in CAO_rapid events. For the effect of the diabatic process, a much stronger diurnal variation is observed. This may be related to the relatively small potential temperature lapse rate and NHC in CAO_slow events (Figure 11 and Table 2). Strong diabatic heating also shows a notable contribution to the cold airmass reduction during the daytime, although the daily average effect of the diabatic process is relatively small. For the decrease in cold airmass associated with the dynamic process, the decreasing rate in CAO_slow events could reach a maximum rate of −116.0 hPa day$^{-1}$, which is 52.8% higher than that in CAO_rapid events. This enhanced contribution of the dynamic process can be related to the strong easterlies in the cold airmass, which are illustrated in the bar charts of Figure 10. The strong eastward transport from NEC to the marine area quickly reduced the cold airmass depth in the CAO_slow events.

## 5 Discussion: large-scale circulation pattern of the CAO associated with the rapid reappearance of air pollution

In this section, we further check the large-scale circulation pattern of the CAO to see if we can find precursors for rapid air pollution reappearance. Figure 14 shows the composited spatial evolutions of cold airmass depth and flux during the initial stages (days −4 to 0) of CAO_rapid and CAO_slow events. For CAO_rapid events (Figure 14a), an anticyclonic pattern of anomalous cold airmass flux appears in northern Siberia (60°–80°N, 70°–120°E) on day −4. To the south of this anomalous anticyclone near Lake Baikal, the cold airmass depth shows a positive anomaly. In the following two days, the anticyclone intensifies in strength and extends from the high latitudes to the middle latitudes covering most of Siberia. The strong southward flux of cold airmass on the eastern flank of the anticyclone induces an increase in cold airmass depth and guides the cold airmass to move southeastward. From day −1 to day 0, the anticyclonic pattern gradually disappears, while the strong southward flux on its eastern flank is strengthened and extends further south. Driven by this southward cold airmass flux, the cold airmass depth anomaly dominates NEC.

The CAO_slow events have distinct features in circulation patterns different from CAO_rapid events (Figure 14b). On day −4, a cyclonic pattern of anomalous cold airmass flux is observed in northeastern Asia, which is also featured by the positive anomaly of cold airmass depth. Meanwhile, a southward flux of cold airmass with cyclonic curvature can be found to the northwest of Lake Baikal (55°–70°N, 80°–100°E). During days −3 and −2, the cyclonic pattern and corresponding cold airmass

depth anomaly over northeastern Asia weakens and moves to the Northwest Pacific. The southward cold airmass flux near Lake Baikal intensifies rapidly and forms a new cyclonic pattern in northeastern Asia. Along with the strengthening of the new cyclonic pattern, a positive anomaly of cold airmass depth occurs and increases rapidly. In the following days, the newly formed cyclonic pattern and anomalous cold airmass depth continued to intensify and swept into NEC.

The above results suggest that CAO_rapid and CAO_slow events are distinguished by the dominant pattern of large-scale
anomalous cold airmass flux in the early period of the CAO (days −4 to −2). The CAO_rapid events feature an anticyclonic pattern in northern Siberia, while CAO_slow events are dominated by a cyclonic pattern in northeastern Asia (Figure 13). The circulation patterns of CAO_rapid and CAO_slow events are also reported as the two main types of CAOs in East Asia, which account for 27% and 26% of the total frequency, respectively (Liu et al., 2021). Such differences in large-scale circulation patterns have a close association with the evolution of cold airmass after CAO. For CAO_rapid events, the southward flux at
the eastern flank of the anticyclone is favorable for the maintenance of cold airmass in NEC (Figures 5, 13a and 14a). For CAO_slow events, however, the cold airmass flux at the southern flank of the cyclone contains a large eastward component, causing a fast departure of cold airmass from NEC (Figures 6, 13b and 14b). Therefore, the anticyclonic and cyclonic patterns in high-latitude Eurasia can be used as precursors for the estimation of air pollution reappearance after the CAO.

**6 Conclusions**

The CAO is well known to have a strong removal effect on air pollution over NEC. However, air pollution may experience a rapid reappearance after the CAO and even lead to worse pollution. Based on the AQI measurement and quantitative description of cold airmass during the last 8 winters, the detailed characteristics and possible causes of air pollution reappearance after 52 CAOs are analyzed. During the period after the CAO, the increase in the AQI in NEC has a probability of more than 80%, and the AQI increase rate is much faster than that in other periods in winter. In more than half of CAOs,
the AQI in reappeared air pollution is worse than the AQI before CAO. The CAOs are divided into a "rapid reappearance" group (63.5%) and a "slow reappearance" group (36.5%) by the increase rate in the AQI. CAO_rapid events tend to have a relatively slow decrease in cold airmass depth and a long interval time before the next CAO. Spatiotemporal evolution shows that the reappearance of air pollution usually occurs in the first two days after CAO. The region of worst AQI after the air pollution reappearance features a slowly decreasing residual cold airmass with nearly no horizontal flux.

The rapid reappearance of air pollution is mainly attributed to the stable stratification in the lower troposphere forced by the intrusion of cold airmass. A comparison of the two groups of CAOs shows that the depth, NHC and vertical structure of cold airmass are key features modulating the reappearance of air pollution. The cold airmass depth could well present the MLH and has good agreement with the MLH in both value and variation. A moderate depth (150–180 hPa) of cold airmass is most favorable for air pollution reappearance since it is thin enough to restrict the spread of air pollutants and thick enough to
maintain its own existence. The NHC is proportional to the vertical stability and persistence of the cold airmass. For a fixed cold airmass depth, a larger NHC will lead to a more rapid reappearance of air pollution. The vertical structure, represented

by the slope of isentropes, in cold airmass could also influence the low-level vertical stability. The cold airmass with relatively flat isentropes will likely have a large vertical component of the potential temperature gradient, which is conducive to the reappearance of air pollution.

Further analysis shows that the evolution of the residual cold airmass and the above conditions are mainly associated with the dynamic process rather than diabatic heating/cooling. The strength of the eastward cold airmass flux after CAO is an important factor affecting the decreasing rate of residual cold airmass. This study also investigates the possible indication of large-scale circulation pattern of the CAO during its initial stage in estimating the reappearance of air pollution after CAO. The CAOs driven by the circulation of anticyclonic pattern in northern Siberia (cyclonic pattern in northeastern Asia) usually leads to a

rapid (slow) deterioration of air pollution.

*Data availability.* The air quality index (AQI) are available at the Ministry of Ecology and Environment of the People Republic of China (https://air.cnemc.cn:18007/). The JRA-55 reanalysis data is available at Japan Meteorological Agency (https://jra.kishou.go.jp/JRA-55). The sounding data are available at University of Wyoming (http://www.weather.uwyo.edu/upperair/sounding.html).

*Author contributions.* QL and GC design of the research, LS and TI provided valuable suggestions. QL downloaded and analyzed the reanalysis and observation data and prepared all the figures. QL wrote the manuscript with contributions from GC, LS and TI.

*Competing interests.* The authors declare that they have no conflict of interest.

*Acknowledgements.* The authors thank the reviewers for their helpful comments and suggestions which greatly improved this

paper. The authors also express thanks for the financial support from Guangdong Basic and Applied Basic Research Foundation and the National Natural Science Foundation of China.

*Financial support.* This work was supported by the Guangdong Major Project of Basic and Applied Basic Research (2020B0301030004) and the National Natural Science Foundation of China (NSFC) (Grant 41805122).

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

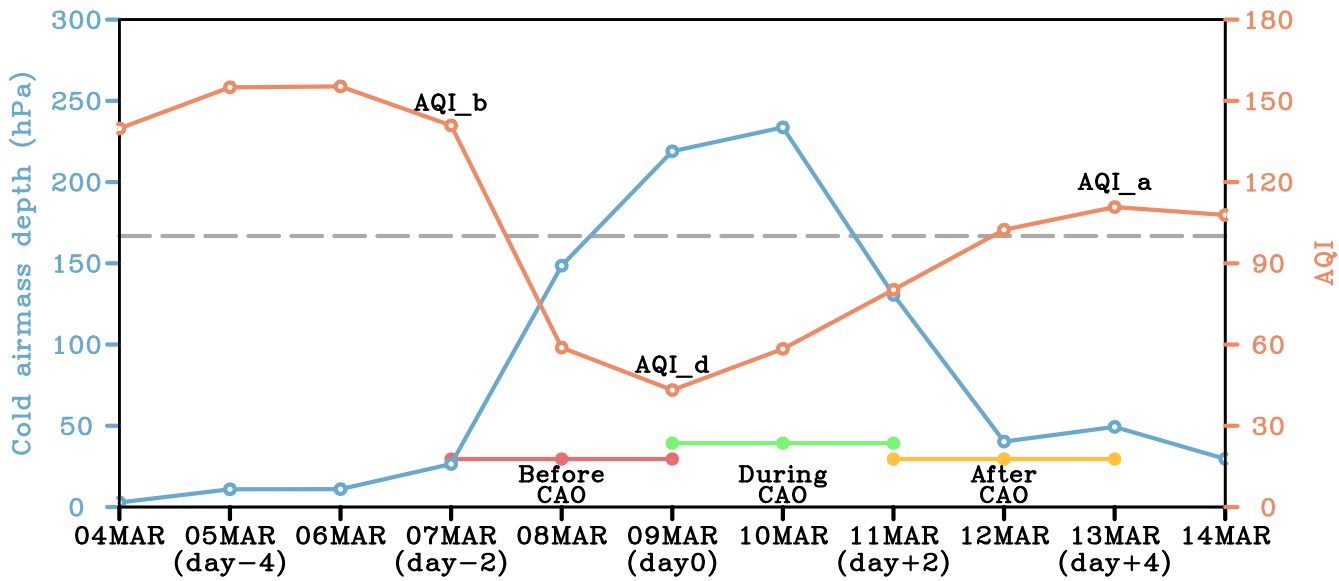

**Figure 1: Time series of the regional averaged cold airmass depth (blue line) and AQI (orange line) in NEC (114°–122°E, 30°–40°N). The gray dashed line denotes the threshold value of the cold airmass depth.**


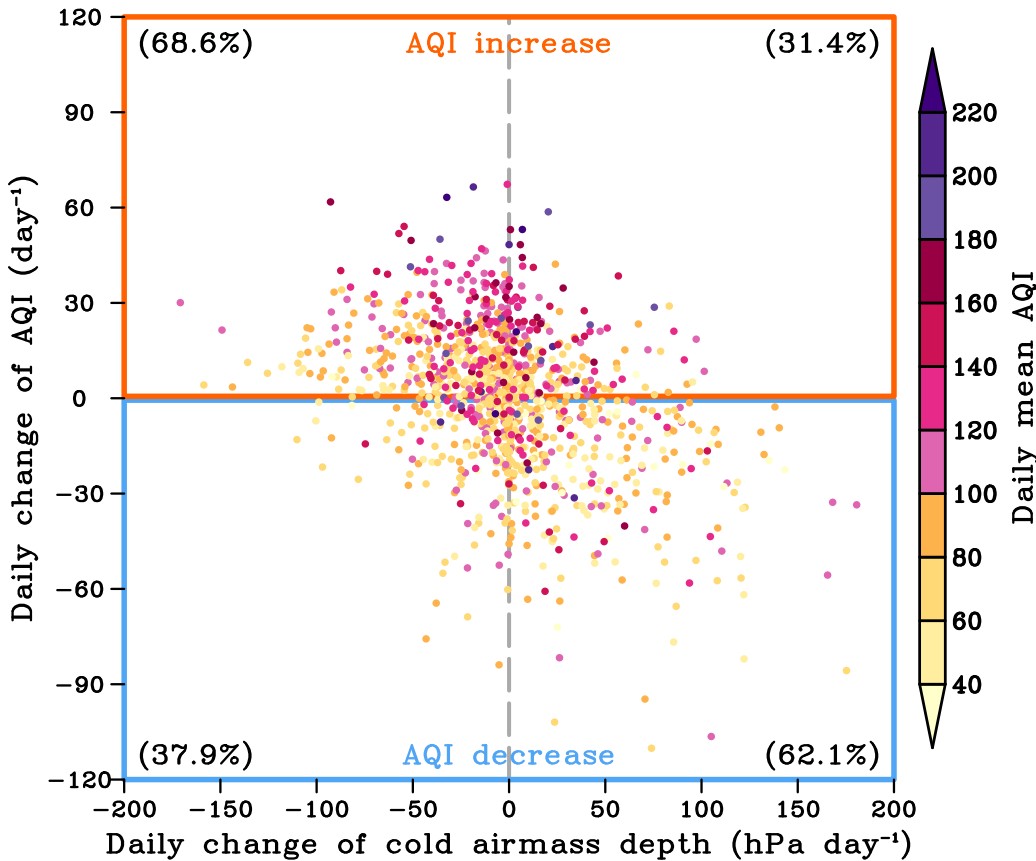

**Figure 2: Relationship between the daily changes in the AQI and cold airmass depth. The daily change is estimated by the current day minus the preceding day. The colors of the dots denote the daily mean cold airmass depth on the current day.**

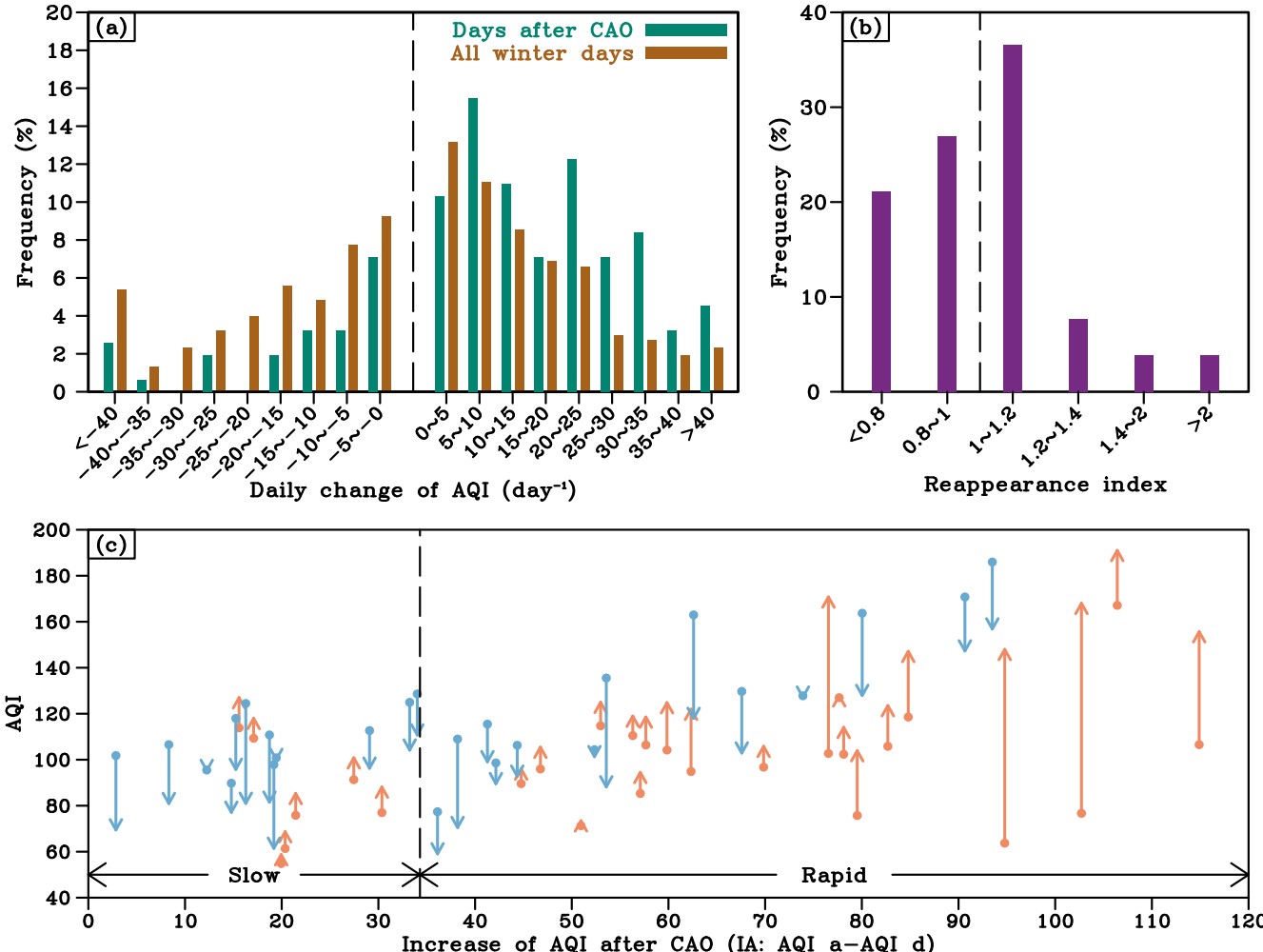

Figure 3: (a) Statistical distribution of daily change of AQI calculated by all of days during the 8 winters (brown bars) and days during the period after the CAOs (green bars). (b) Statistical distribution of the reappearance index of air pollution in 52 CAO events. (c) The increase in the AQI (x-axis) and the change in the AQI from the period before the CAO to the period after CAO (vertical arrows) in 52 CAO events. Black dashed line denotes the standard deviation of AQI. The arrowhead and tail represent AQI_a and AQI_b, respectively. The orange (blue) arrows denote that the AQI worsens (better) after CAO.

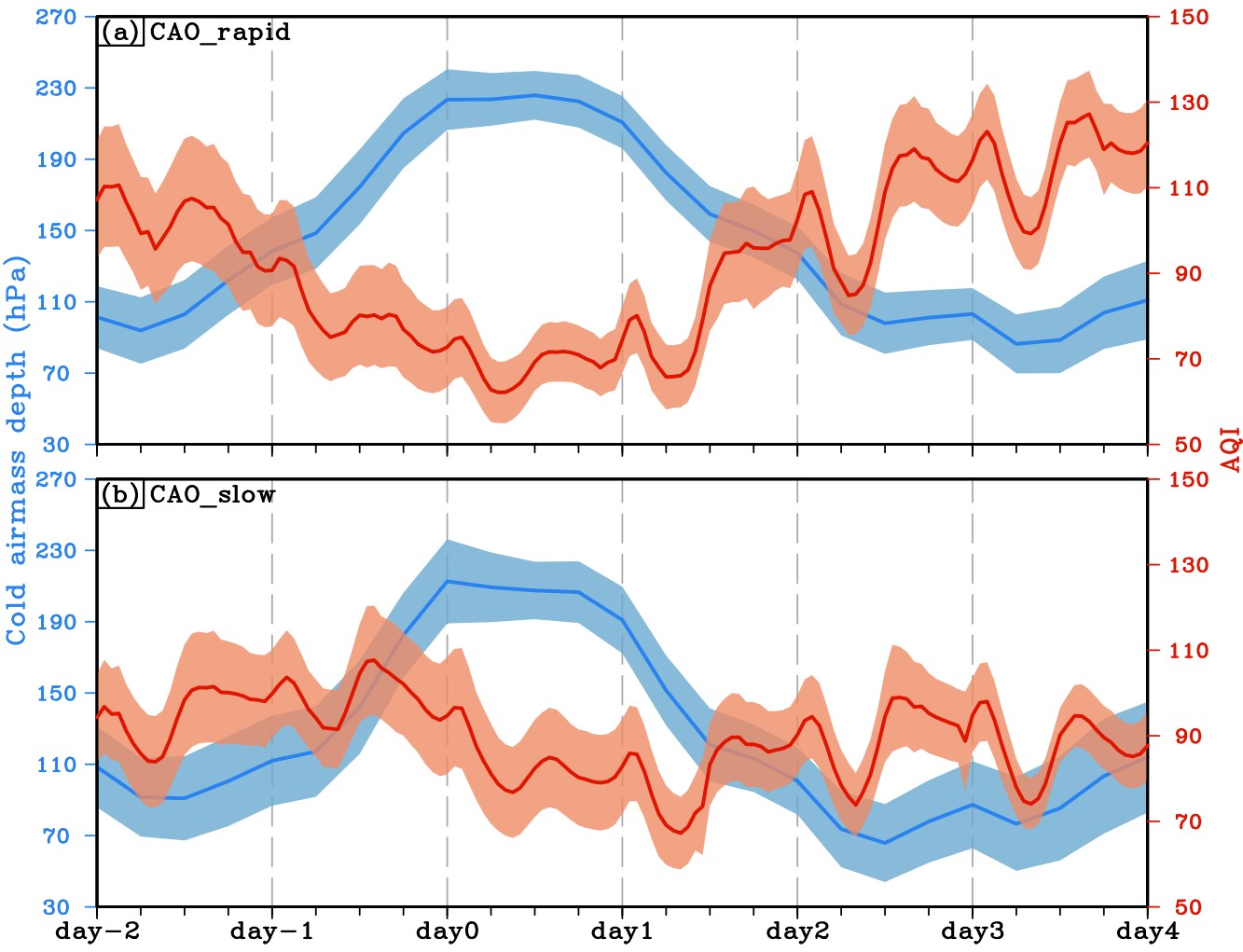


Figure 4: Evolutions of spatially averaged (NEC: 114°–122°E, 30°–40°N) cold airmass depth and AQI during CAOs. (a) and (b) are composited by the CAO_rapid events and CAO_slow events, respectively. Blue lines and red lines denote the cold airmass depth and AQI, respectively. Shading represents the 95% confidence interval of the composited mean value.

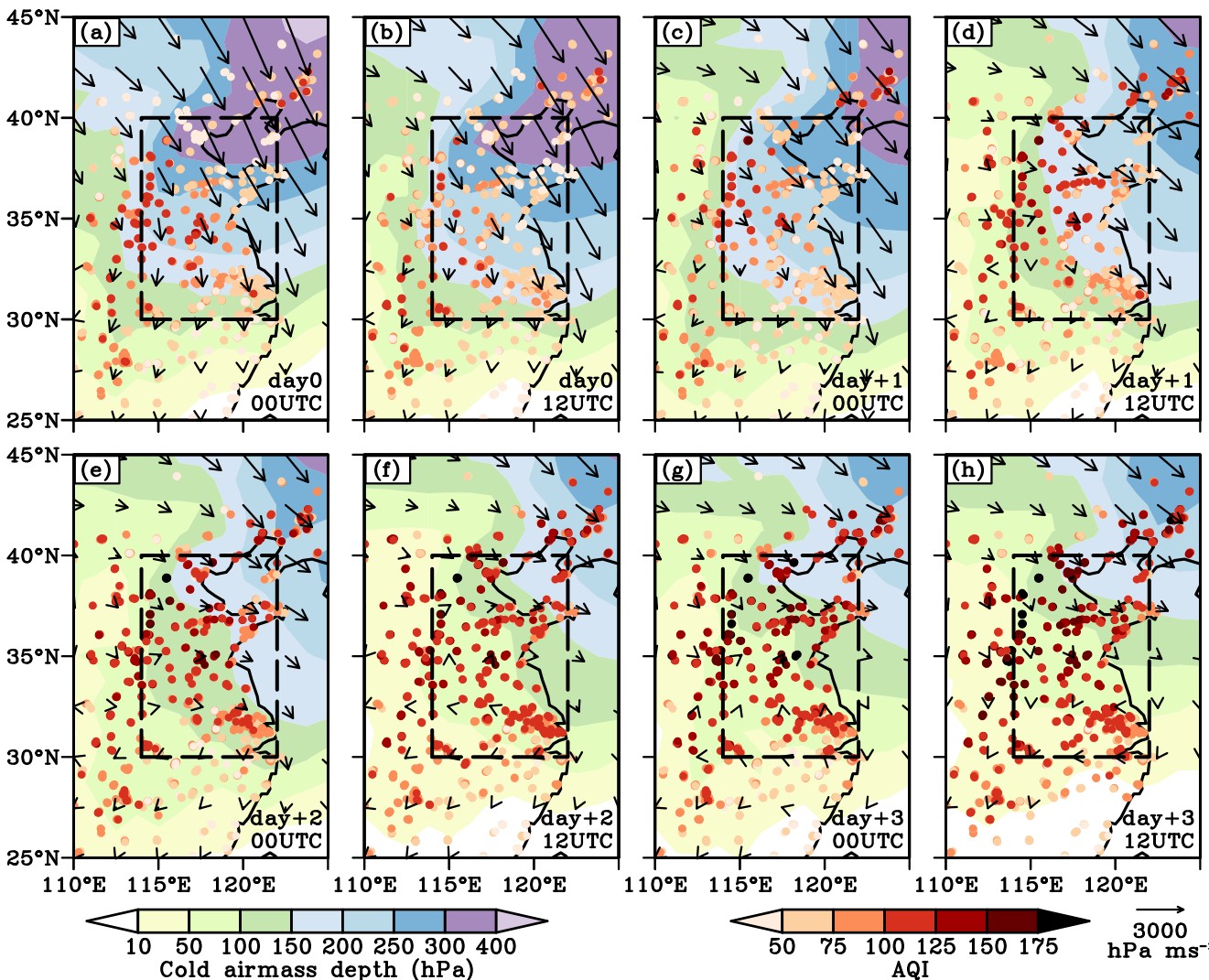

**Figure 5: Spatial distributions of the composited cold airmass depth (shaded), horizontal fluxes of the cold airmass (black vectors), and AQI (colored dots) during CAO_rapid events. The black boxes denote the NEC (114°–122°E, 30°–40°N).**

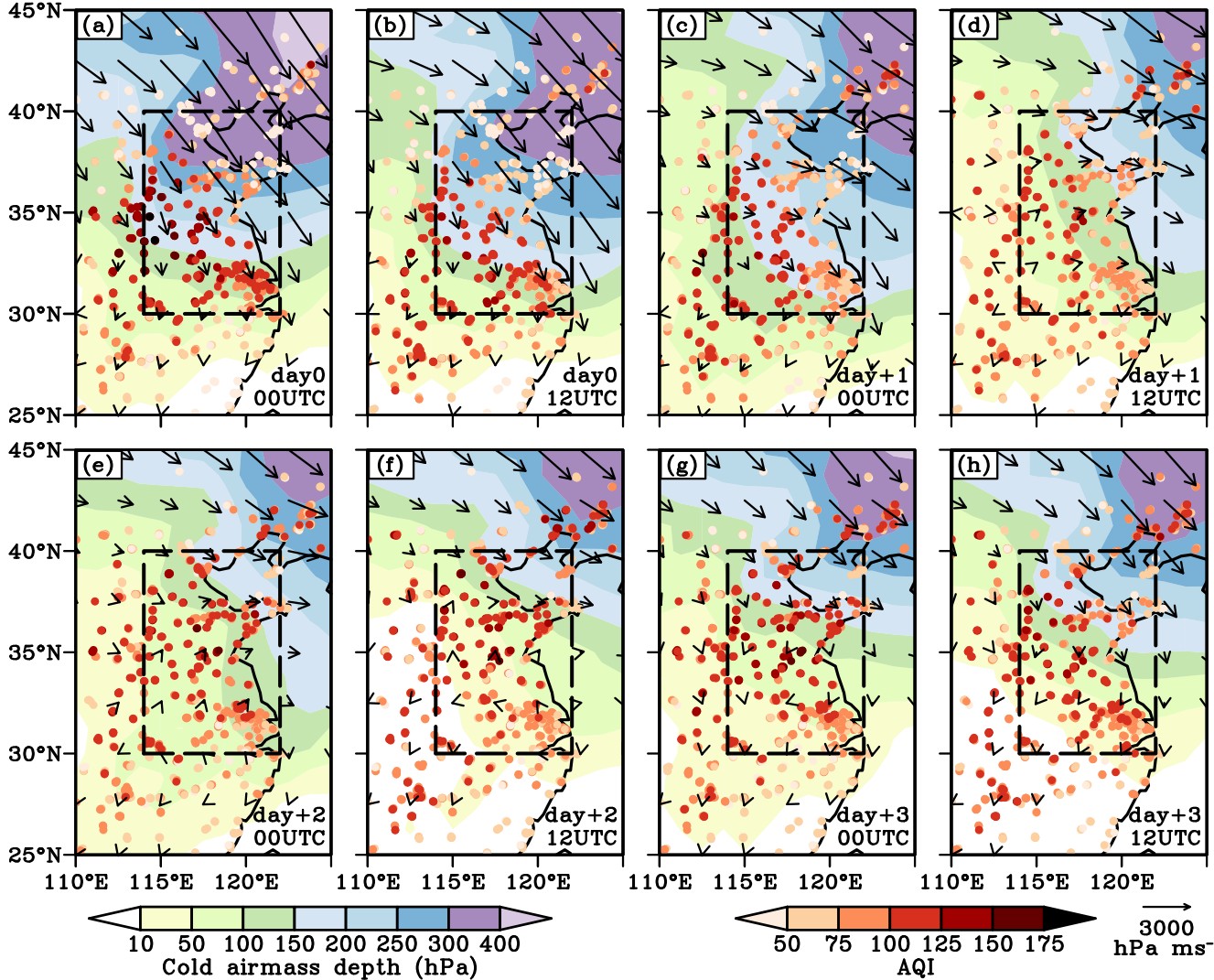

**Figure 6:** The same as Figure 5 but composited by the CAO_slow events.

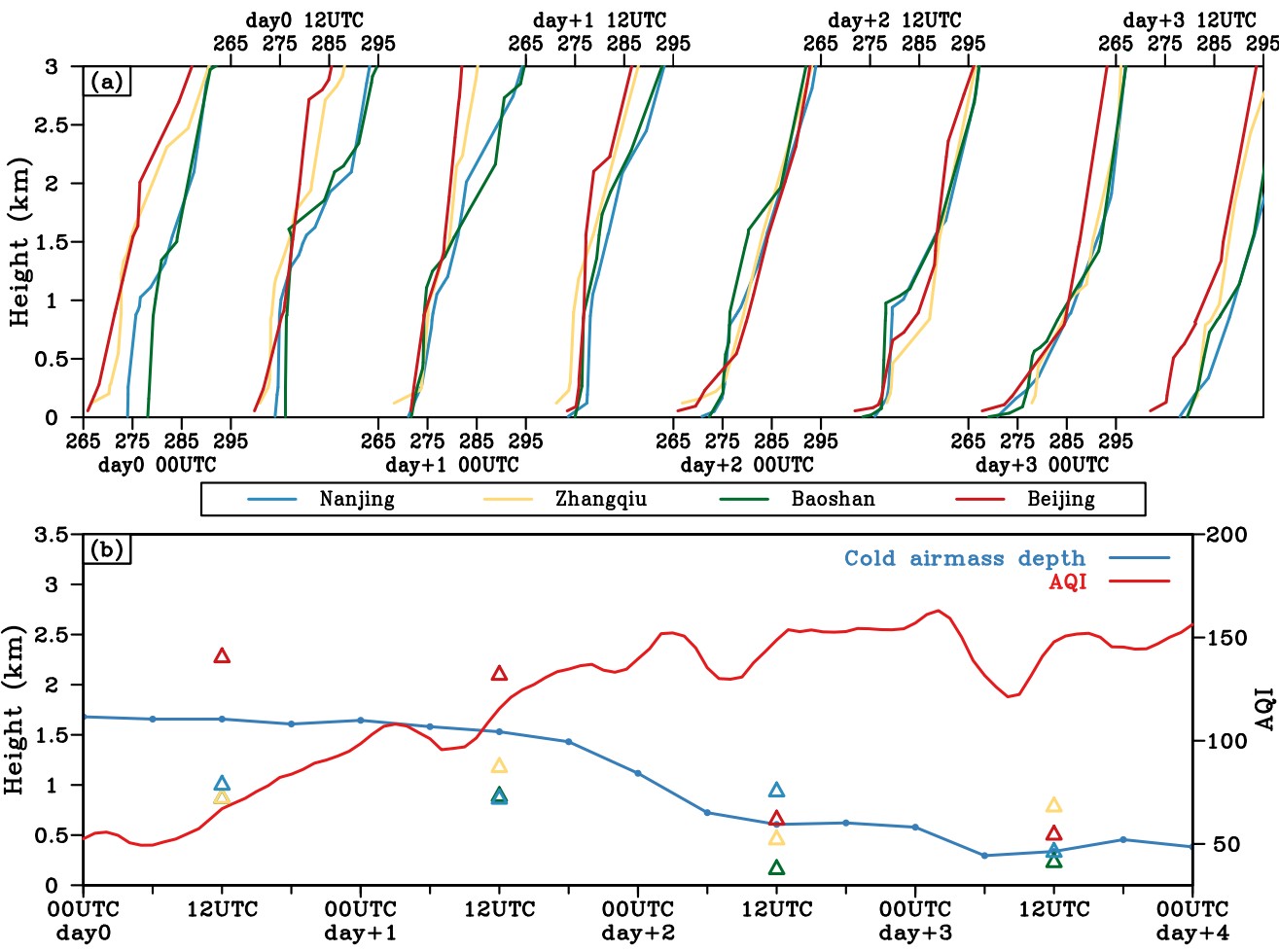

Figure 7: (a) Vertical profiles of potential temperature at four stations (from north to south: Beijing, Zhangqiu, Nanjing, Baoshan) during a CAO rapid event from 14 to 17 Dec 2016. (b) Time series of the regional averaged cold airmass depth and the AQI in NEC and MLH (colored triangles) at four stations during a CAO_rapid event from 14 to 17 Dec 2016.


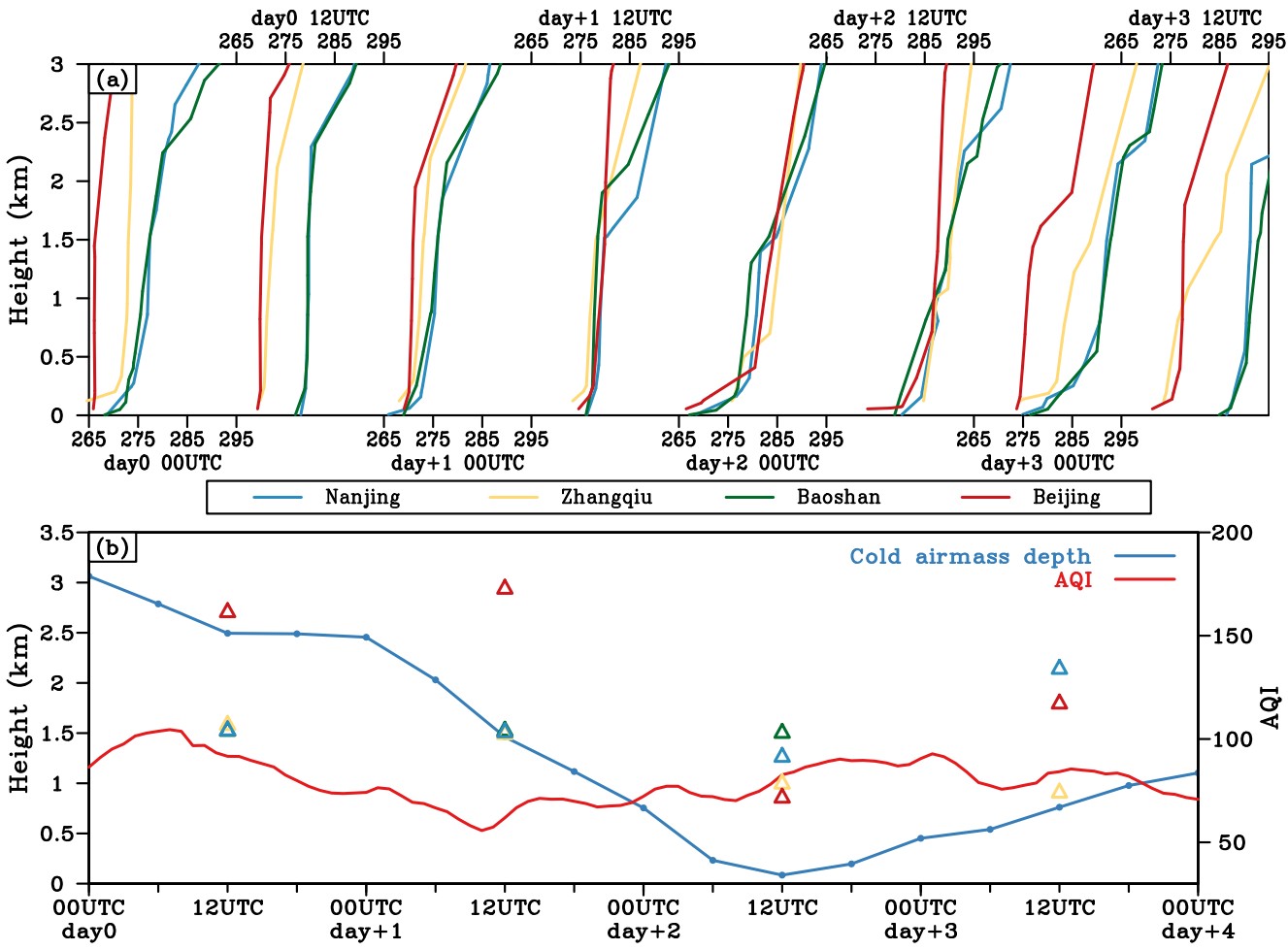

Figure 8: The same as Figure 7 but during a CAO_slow event from 11 to 14 Feb 2018.

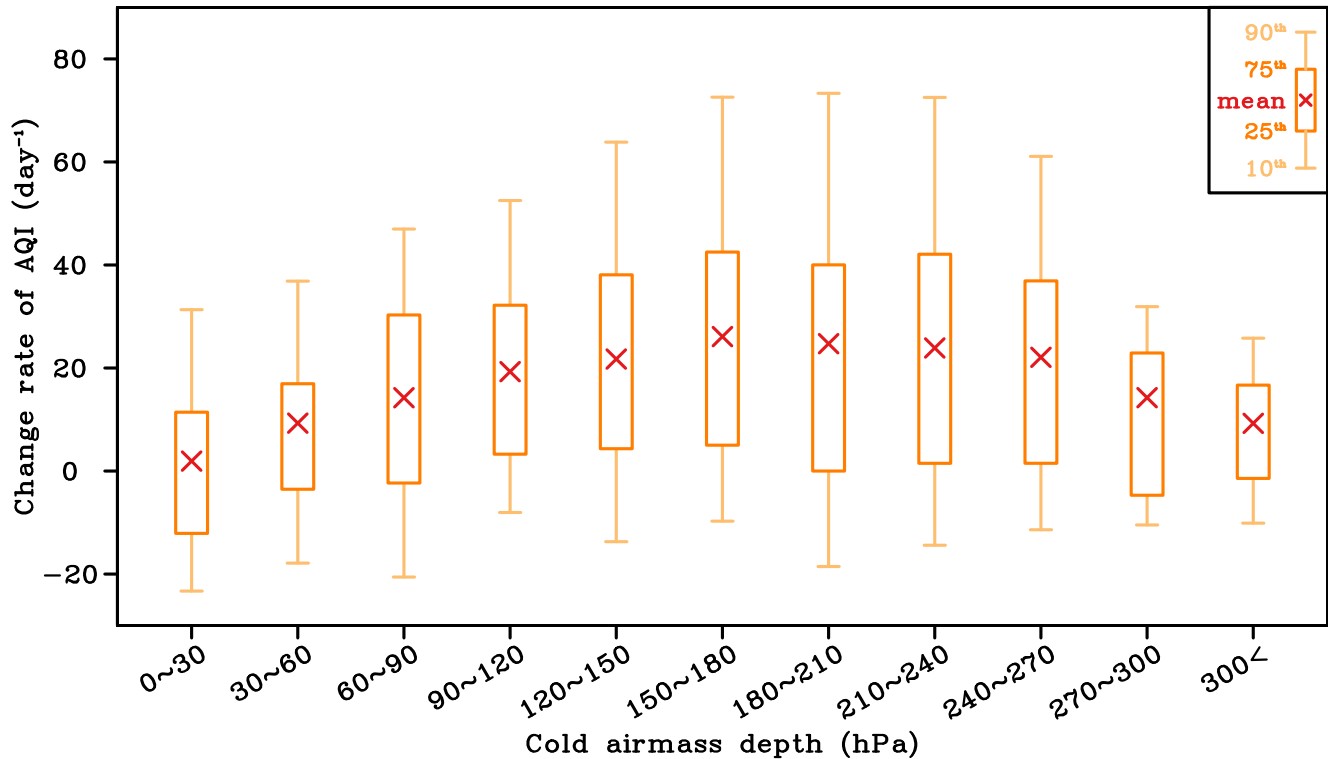


**Figure 9: Distribution of the change rate of the AQI with the cold airmass depth during the period after the CAO.**

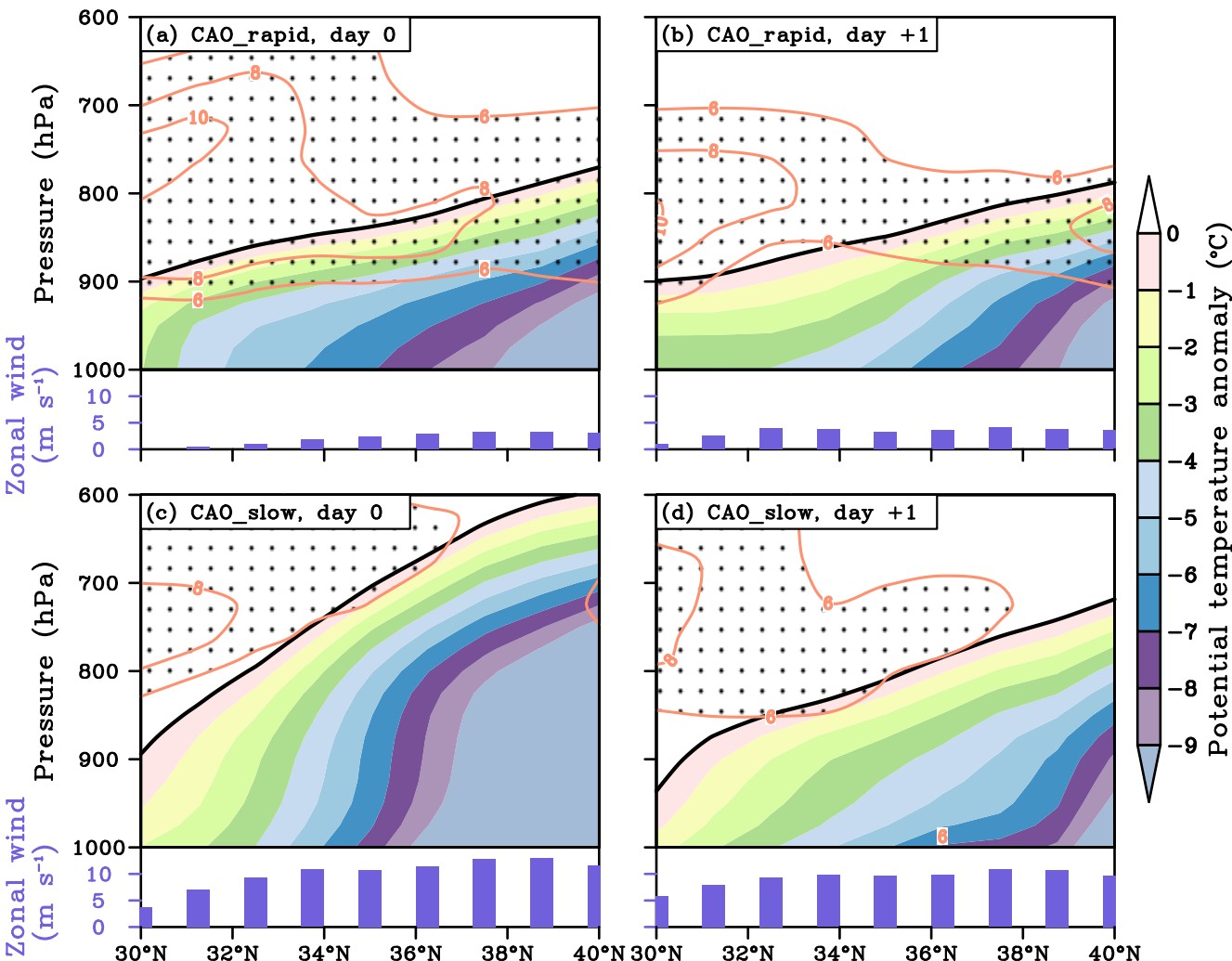

**Figure 10: (a, b)** Longitude-vertical sections of the potential temperature anomaly (colored shading) and vertical gradient of the potential temperature (contour, unit: $10^{-3} \, K \, m^{-1}$, dotted for $> 6 \times 10^{-3} \, K \, m^{-1}$) along 119°E during the CAO_rapid event from 14 to 17 Dec 2016. The thick black line denotes the upper boundary of cold airmass. The purple bars denote the mean zonal wind speed in the cold airmass. **(c, d)** The same as (a, b) but during the CAO_slow event from 11 to 14 Feb 2018.

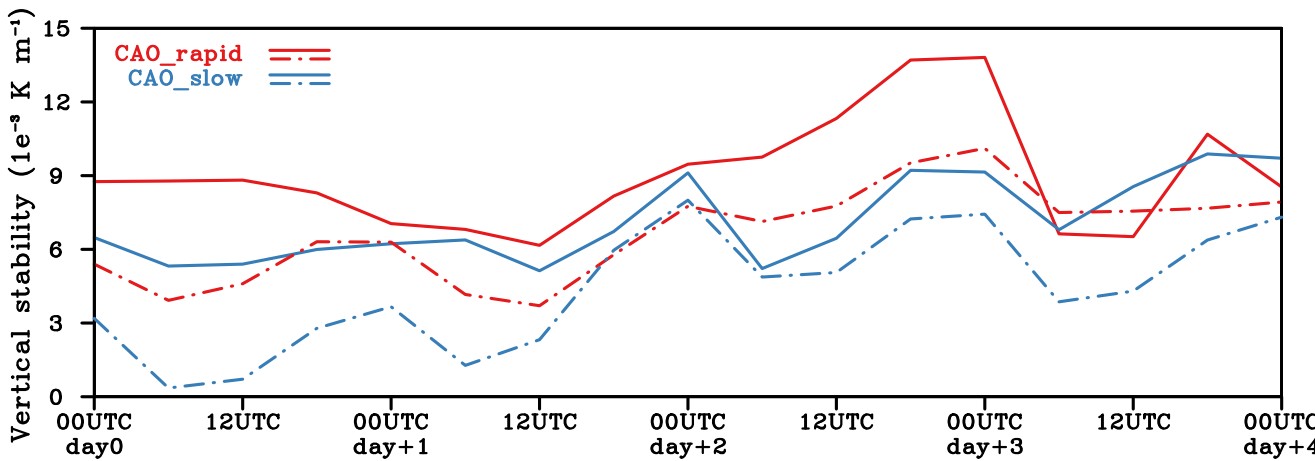

**Figure 11: Evolutions of spatially averaged (NEC: 114°–122°E, 30°–40°N) vertical stability at the upper boundary of cold airmass (solid line) and the averaged vertical stability in the cold airmass (dash dot line) during a CAO_rapid event from 14 to 17 Dec 2016 (red) and a CAO_slow event from 11 to 14 Feb 2018 (blue).**

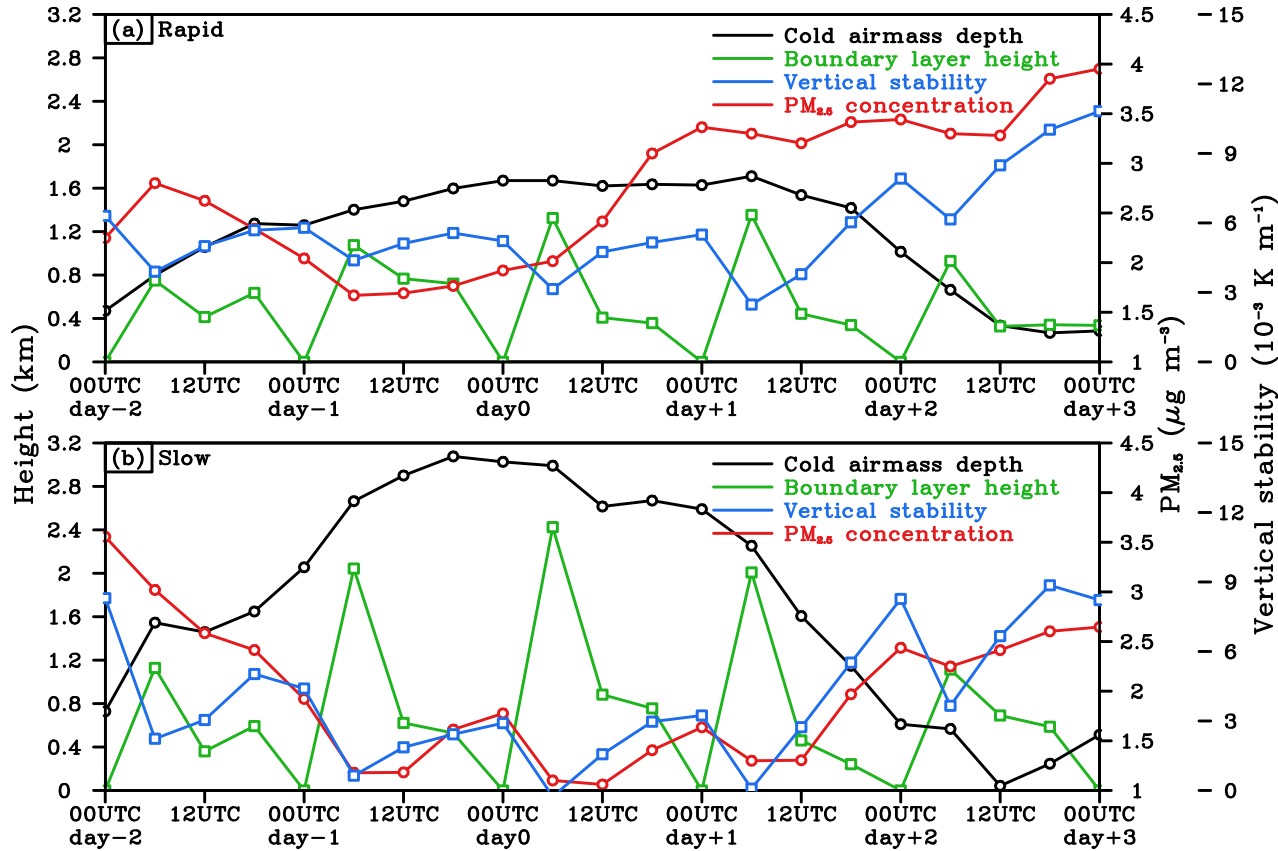

**Figure 12: Evolutions of spatially averaged (NEC: 114°–122°E, 30°–40°N) cold airmass depth, atmospheric boundary layer height, vertical stability (1000–850 hPa) and surface PM2.5 concentration during (a) CAO_rapid event from 14 to 17 Dec 2016 and (b) CAO_slow event from 11 to 14 Feb 2018.**

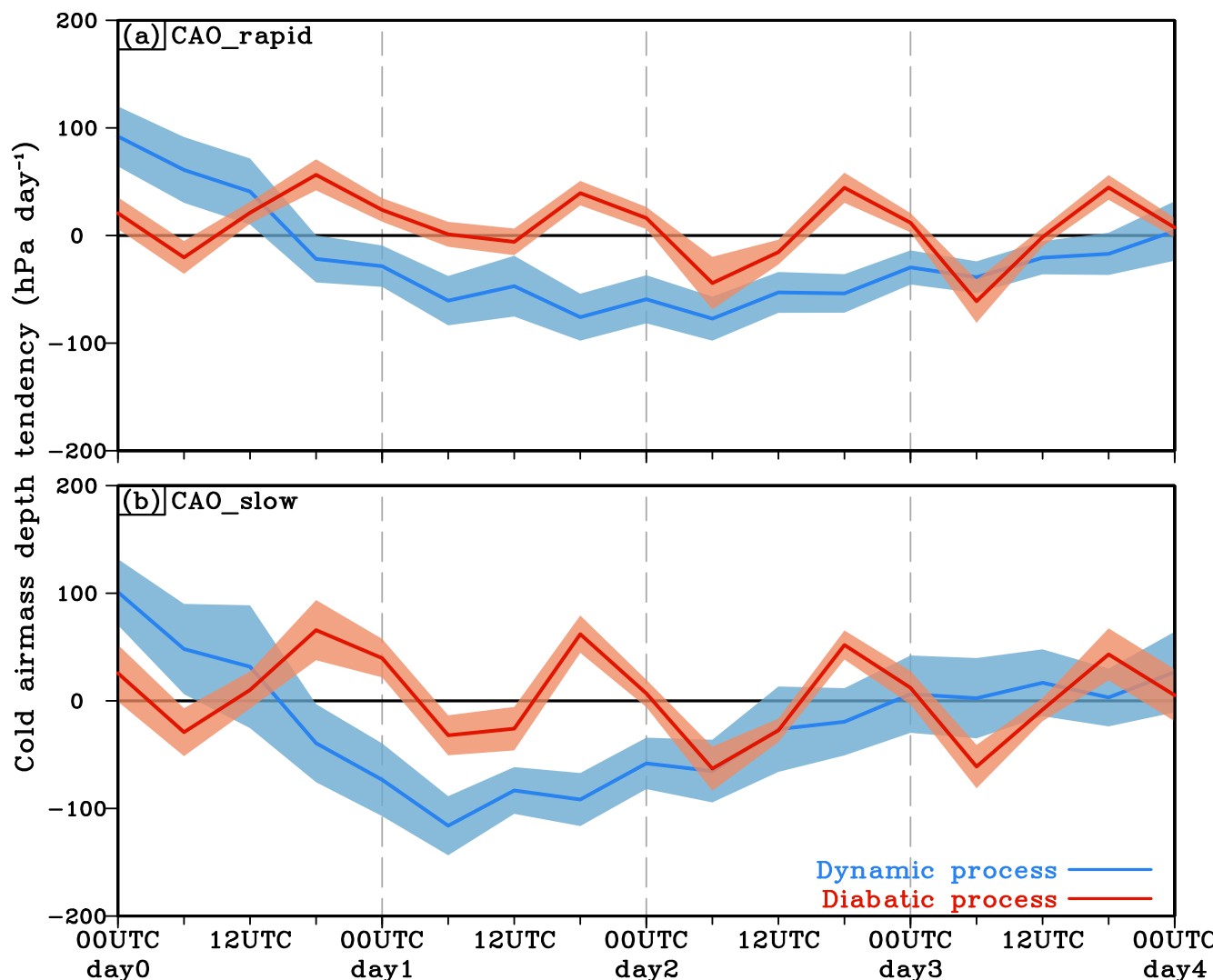

**Figure 13: Evolutions of the spatially averaged (NEC: 114°–122°E, 30°–40°N) cold airmass depth tendency during CAOs. Blue and red lines denote the contributions of dynamic and diabatic processes to the tendency of cold airmass depth, respectively. (a) and (b) are composited by the CAO_rapid events and CAO_slow events, respectively. Shading represents the 95% confidence interval of the composited mean value.**


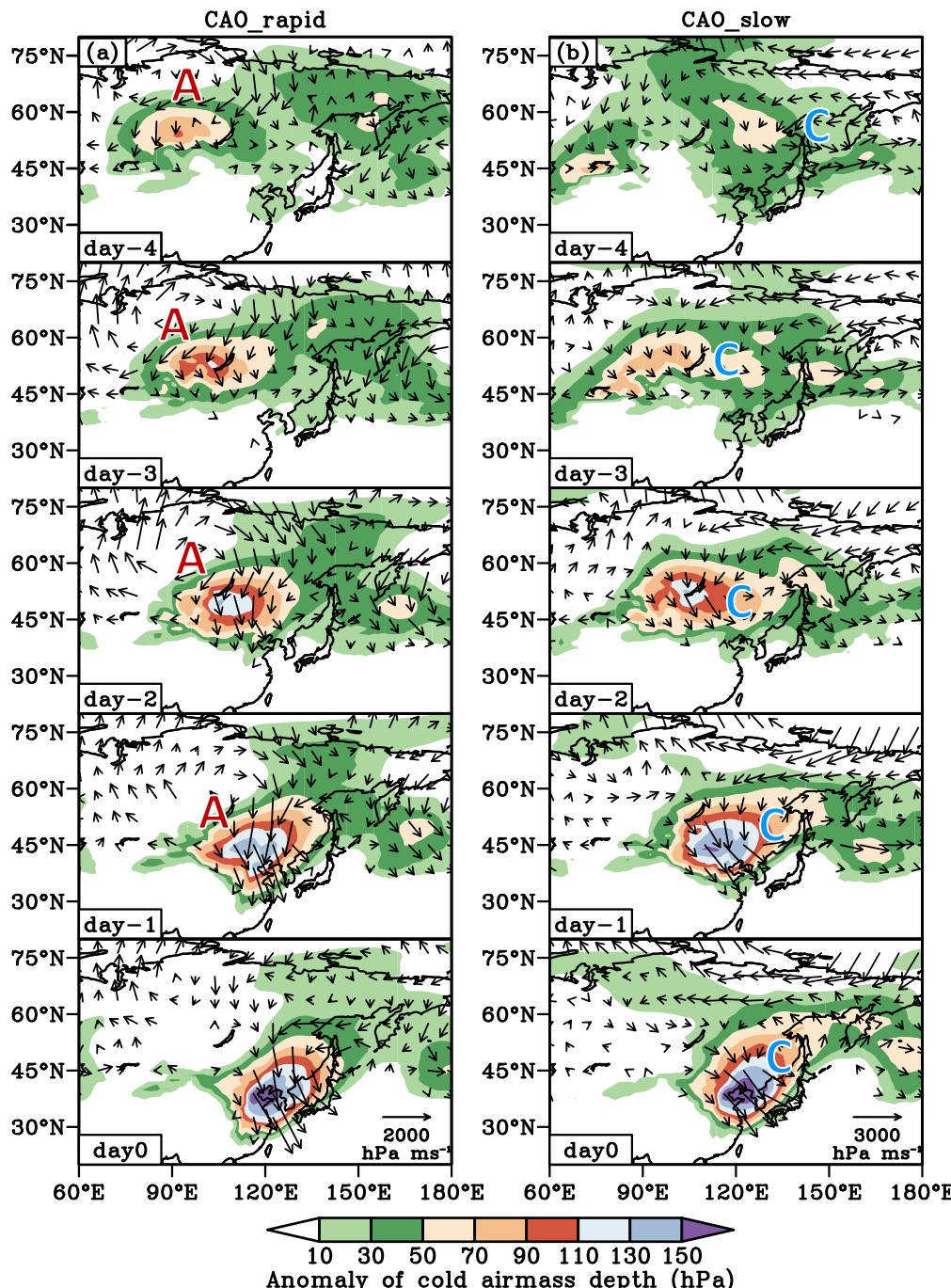

**Figure 14: Large-scale anomalies of cold airmass depth (shading) and cold airmass flux (vectors) in the initial stage from day −4 to 0. (a) and (b) are composited by the CAO_rapid events and CAO_slow events, respectively.**