# Peer review of "Rapid reappearance of air pollution after cold air outbreaks in northern and eastern China"

_Atmospheric Chemistry and Physics, 2022_

## Author Comment (AC1)

\*\*\*\*\*\*\*\*\*\*\*\*\*\*\*\*\*\*\*\*\*\*\*\*\*\*\*\*\*\*\*\*\*\*\*\*\*\*\*\*\*\*\*\*\*\*\*\*\*\*\*\*\*\*\*\*\*\*\*\*\*\*\*\*\*\*\*\*\*\*\*\*\*\*\*\*

In this file, **the text in black** shows the comments from reviewers and editor, while **the text in blue** is our replies.

\*\*\*\*\*\*\*\*\*\*\*\*\*\*\*\*\*\*\*\*\*\*\*\*\*\*\*\*\*\*\*\*\*\*\*\*\*\*\*\*\*\*\*\*\*\*\*\*\*\*\*\*\*\*\*\*\*\*\*\*\*\*\*\*\*\*\*\*\*\*\*\*\*\*\*\*

**SUMMARY:**

Using observation data and reanalysis data, this paper utilizes a quantitative measurement of cold airmass to identify CAO and related dynamic/thermodynamic properties. The authors find the generic existence of air pollution reappearance after CAO, and raise a possible mechanism in the manuscript. This manuscript overall is interesting, but I do have some comments regarding the details of data and methods. My major and minor concerns are described below.

**Response:** Thank you very much for your kind evaluation. All of your comments are accepted and revised accordingly. We consider that your comments have helped greatly to improve the manuscript. The detailed response and revision are given below.

**MAJOR COMMENTS:**

1. The authors define north China as 30-40 N, 114-122 E. However, this is not a good definition. Furthermore, two sounding stations used in this manuscript, Nanjing and Baoshan (which is in Shanghai), are in East China. The authors can refer to the definition of the North China Plain in Kang et al. 2018

Kang, S., Eltahir, E.A.B. North China Plain threatened by deadly heatwaves due to climate change and irrigation. Nat Commun. 9, 2894,2018. https://doi.org/10.1038/s41467-018-05252-y

**Response:** We agree that the spatial range of North China Plain is usually defined as 113°–121°E, 34°–41°N (Kang and Eltahir, 2018), which is slightly north than the definition in our study. Here, we would like to explain why we select the region 114°–122°E, 30°–40°N as the study area. The determination of study area is based on the characteristics of air pollution and cold air activity. Figure R1 shows that the air pollution (AQI > 100) mainly occurs between the latitudes of 30°N and 40°N. The cold airmass activities also can affect areas around 30°N. In addition, our previous study shows the air quality at 30°N (Nanjing and Shanghai) usually varies following (only a half day's delay) the changes of air quality at 40°N during a CAO (Figure 7 in Liu et al., 2019). This suggest the changes of air quality in area between 30°N and 40°N could be considered as a whole.

In addition, Figure 1 in Kang and Eltahir (2018) shows the spatial distribution of topography, area equipped for irrigation, highest daily maximum wet-bulb temperature and population density. The characteristics of these variables, as well as AQI and cold airmass depth in Figure R1, in areas 30°–34°N and 34°–40°N are highly consistent, indicating the area 30°–40°N (Black boxes) is more appropriate to be selected as the study region.

Following your suggestion, we refer to the study region (114°–122°E, 30°–40°N) as "Northern and Eastern China" (NEC). We also add some explanations of the study region in Section 2 at Lines 85–88: "The determination of study area (NEC), including parts of both northern China and eastern China, is based on the characteristics of both air pollution and cold

air activity. The air pollution (AQI > 100) mainly occurs between the latitudes of 30°N and 40°N, and the CAOs also usually affect areas around 30°N (figure omitted)."

[Figure]

**Figure R1**. Winter mean cold airmass depth (shading) and AQI (colored dots) during 2014/2015–2021/2022.

[Figure]

**Figure 1 in Kang and Eltahir 2018**. Brief characterization of Eastern China. Spatial distribution of a topographic map (m), b area equipped for irrigation (AEI, %) for 2005 from Historical Irrigation Data with climatology of annual precipitation from TRMM (contour, mm) in modern record (1998–2015), c highest daily maximum wet-bulb temperature from ERA-Interim, $TW_{max}$ (°C) in modern record (1979–2016), and d population density in people/km². The box in each plot indicates the North China Plain used for regional analysis in this study.

**References**

Liu Q, Chen G and Iwasaki T 2019 Quantifying the impacts of cold airmass on aerosol concentrations over North China using isentropic analysis *J. Geophys. Res. Atmos.* **124** 7308–7326

2. In Section 2.1, more details are needed. For example

(1) What's the spatial distribution of those local AQI stations? Are most of them in the big cities?

**Response:** Figure R2 shows the spatial distribution of AQI stations used in our study. The stations have a relative even distribution in the study area. Some of the AQI stations are located in big cities (shaded by dark red) and some of them are in rural/suburban region (shaded by light yellow). According to your comments, we add description about the AQI observations at Lines 88–89: "The AQI stations have a relative even distribution in the study area (figure omitted). The stations are located in not only urban areas but also suburban and rural areas."

[Figure]

**Figure R2**. Spatial distribution of AQI stations (green circles) and nighttime light (shading) in 2013. The nighttime light values range from 0-63 and large (small) value denotes the urban (rural/suburban). The white box denotes the northern and eastern China. (Data source https://www.ngdc.noaa.gov/eog/dmsp.html).

(2) What's the vertical and horizontal resolution for the sounding data? And why do authors use sounding station data for the wind and air temperature? The temporal resolution is not good. Why don't you use reanalysis data as well?

**Response:** The sounding data used in our study has a higher vertical resolution than reanalysis data especially in atmospheric boundary layer. For example, sounding data at Beijing station has 60–70 levels in total and 8–14 levels below 850 hPa during study period of 14–17 Dec 2016 (Figure 7a in manuscript), while the reanalysis data has 37 vertical levels with 7 levels below 850 hPa (see reply to MAJOR COMMENT #2(3)). For spatial resolution, there are not many available radiosonde stations in the study area. So, we only selected four stations distributed from north to south. We add some of above descriptions of sounding data at Lines 95–97: "The sounding data used in our study usually has a higher vertical resolution than mainstream reanalysis data especially in atmospheric boundary layer. For example, sounding data at Beijing station has 60–70 levels in total and 8–14 levels below 850 hPa during a CAO event of 14–17 Dec 2016."

The sounding data provides a direct detection of atmospheric vertical profiles, which is more reliable than the model-simulated reanalysis data. Some studies also show that the boundary layer height calculated by reanalysis data has some biases as compared to sounding observation (Guo *et al*., 2021). Moreover, the atmospheric boundary layer conditions calculated by sounding observation data could well explain the changes of AQI observations. Therefore, we need first to verify the reliability of reanalysis data with sounding data.

We agree with the reviewer's suggestion that reanalysis data may be a good substitute for sounding data, since reanalysis data has a higher spatial and temporal resolution. Indeed, we have used reanalysis data to analyze the boundary layer characteristics. In section 4.1, we found the depth of residual cold airmass calculated by reanalysis data is highly consistent with the mixing layer height calculated by sounding data. The depth of residual cold airmass and mixing layer height also has a physical connection as discussed in the manuscript at Lines 233–243. This result suggests the use of reanalysis data is appropriate and reliable in our study. In addition, the calculation of cold airmass depth using reanalysis data is much easier than the calculation of MLH using sounding. Therefore, we used the reanalysis data instead of sounding data in the rest part of section 4.

**References**

Guo J, Zhang J, Yang K, Liao H, Zhang S, Huang K, Lv Y, Shao J, Yu T, Tong B, Li J, Su T, Yim S H L, Stoffelen A, Zhai P and Xu X 2021 Investigation of near-global daytime boundary layer height using high-resolution radiosondes: first results and comparison with ERA5, MERRA-2, JRA-55, and NCEP-2 reanalyses *Atmos. Chem. Phys.* **21** 17079–17097

(3) I am not sure which JRA-55 products are used in this study. First of all, JRA-55 should have a 3-hourly reanalysis, and usually, the vertical pressure levels are 60 levels.

**Response:** In this study, we use the standard product of isobaric analysis data (6-hourly, 37 layers) of JRA-55 following previous studies on cold air outbreak (Kanno *et al*., 2015; Abdillah *et al*., 2017; Liu *et al*., 2021). This data is freely available at JMA (http://search.diasjp.net/en/dataset/JRA55) and NCAR (https://rda.ucar.edu/datasets/ds628.0/).

The JRA-55 product users' handbook can be found in this website (https://jra.kishou.go.jp/JRA-55/document/JRA-55_handbook_LL125_en.pdf). At page 8 of the handbook, the time interval of isobaric analysis data is described as: "These fields are produced every six hours at 00, 06, 12 and 18 UTC". Page 14 shows the vertical coordinates of the data in isobaric fields as follow: "Isobaric fields are produced for 37 isobaric surfaces

(1000, 975, 950, 925, 900, 875, 850, 825, 800, 775, 750, 700, 650, 600, 550, 500, 450, 400, 350, 300, 250, 225, 200, 175, 150, 125, 100, 70, 50, 30, 20, 10, 7, 5, 3, 2 and 1 hPa) except dew-point depression (or deficit), specific humidity, relative humidity, cloud cover, cloud water, cloud liquid water and cloud ice, which are produced for 27 levels from 1000 to 100 hPa only".

Following your suggestion, we add some detailed descriptions at Lines 77–79: "To identify the CAO events, we used isobaric analysis data of Japanese 55-year reanalysis (JRA-55). This dataset is freely available at JMA (http://search.diasjp.net/en/dataset/JRA55) and NCAR (https://rda.ucar.edu/datasets/ds628.0/). The JRA-55 has a horizontal resolution of 1.25° with 37 vertical pressure levels and a time interval of 6 hours (00, 06, 12 and 18 UTC)."

**References**

Kanno Y, Abdillah M R and Iwasaki T 2015 Charge and discharge of polar cold air mass in northern hemispheric winter *Geophys. Res. Lett.* **42** 7187–7193

Abdillah M R, Kanno Y and Iwasaki T 2017 Tropical–extratropical interactions associated with East Asian cold air outbreaks. Part I: Interannual variability *J. Clim.* **30** 2989–3007

Liu Q, Chen G, Wang L, Kanno Y, and Iwasaki T 2021 Southward cold airmass flux of the East Asian winter monsoon: Diversity and impacts *J. Clim.* **34** 3239–3254

**MINOR COMMENTS:**

1. In the introduction, the recent COVID-19 lockdowns also provide a unique opportunity to study the complex chemical effects of air pollution as well as meteorology. Here are some references.

1) Le T, Wang Y, Liu L, Yang J, Yung YL, Li G, Seinfeld JH. Unexpected air pollution with marked emission reductions during the COVID-19 outbreak in China. Science. 2020 Aug 7;369(6504):702-6.

2) Wang Y, Wen Y, Wang Y, Zhang S, Zhang KM, Zheng H, Xing J, Wu Y, Hao J. Four-month changes in air quality during and after the COVID-19 lockdown in six megacities in China. Environmental Science & Technology Letters. 2020 Sep 9;7(11):802-8.

3) Zhao N, Wang G, Li G, Lang J, Zhang H. Air pollution episodes during the COVID-19 outbreak in the Beijing–Tianjin–Hebei region of China: an insight into the transport pathways and source distribution. Environmental Pollution. 2020 Dec 1;267:115617.

**Response:** Following your suggestion, we add above references and some discussions in the introduction at Lines 36–37: "Even when local emission is obviously reduced during COVID-19 lockdown, severe air pollution still occurs in North China (Zhao et al., 2020)." and Lines 40–42: "The recent COVID-19 lockdowns also provide a unique opportunity for studying the complex chemical effects of air pollution as well as meteorology (Le et al., 2020; Wang et al., 2020)."

2. Section 2.1, "with observations made 24 times per day", is it measured equal frequency (i.e., 1h frequency)?

**Response:** Yes, the observation has a time interval of 1 hour. The revised text can be found at Line 84: "with observations of 1hour frequency".

3. Line 87, 89, vertical integral -> vertical integration.

**Response:** Revised as your suggestion.

4. For the definition of mass flux, the common definition is the rate of mass flow (SI unit kg/(m^2 s).

**Response:** We agree the unit of mass flux is usually expressed as kg/(m$^2$ s). In our study, we use the horizontal flux of cold airmass to describe the horizontal movement of cold air, which is different from the mass flux of cold air.

Following Iwasaki et al. (2014), the total cold airmass amount/depth (*DP*) is defined as the depth of air layer between ground surface ($p_s$) the isentropic surface of threshold potential temperature ($\theta_T$), $DP = p_s - p(\theta_T)$ (area below the thick solid contour of 280K in Figure 2 in Iwasaki *et al* 2014). Thus, the definitions of dynamical and thermal properties are based on such definition of cold airmass (*DP*, unit: hPa). $\theta_T$

Here, the isentropic mass continuity equation can be written as

$$\frac{\partial}{\partial t}\left(\frac{\partial p}{\partial \theta}\right) + \nabla \cdot \left(\frac{\partial p}{\partial \theta}v\right) + \frac{\partial}{\partial \theta}\left(\frac{\partial p}{\partial \theta}\dot{\theta}\right) = 0.$$

The vertical integration of above equation from surface boundary ($\theta_s$) to $\theta_T$ leads us to total cold airmass conservation

$$\frac{\partial}{\partial t}DP = -\nabla \cdot \int_{p(\theta_T)}^{p_s} vdp + G(\theta_T),$$

where $\int_{p(\theta_T)}^{p_s} vdp$ is defined as the horizontal flux of cold airmass (*F*). Thus, the cold airmass flux (*F*) has a unit of hPa m s$^{-1}$.

[Figure]

[Figure]

**Figure 2 in Iwasaki *et al* 2014**. Meridional cross sections of MIM's mass stream functions (contour interval $10^{10}$ kg s$^{-1}$) and potential temperature (contour interval 10 K) for DJF means over 1980/81–2009/10. A black dot is placed at the turning point of 45°N and isentropic zonal mean pressure of 850 hPa. Black colors indicate the zonally averaged topography.

5. Line 95, why do authors use the standard deviation to define the CAO? This only makes sense when the mean value of cold airmass depth is very close to zero. (Think about one example, if the cold airmass depth is 50 hPa, the standard deviation is still 169.7 hPa, then CAO should be defined as a cold airmass depth exceeding 50+169.7 = 219.7 hPa

**Response:** Sorry for the misleading description in the original manuscript. We fully agree that the selection of threshold value should consider the sum of standard deviation and mean value, which has been adopted in our study. The standard deviation and mean value of regional mean cold airmass are 78.6 hPa and 91.1 hPa, respectively. Thus, the threshold value of CAO is set as 78.6 + 91.1 = 169.7 hPa.

It should be noted that the study period has expanded to 2014/2015~2021/2022 in the revised manuscript. Thus, the threshold value of CAO becomes 166.8 hPa (i.e. 77.3+89.5 hPa). See revised text at Lines 112–114: "Thus, the CAO in this study is identified when the regional mean cold airmass depth exceeds 166.8 hPa, which is the sum of mean value (77.3 hPa) and standard deviation (89.5 hPa) of cold airmass depth on all winter days. According to the above criteria, 52 CAOs are identified over the 8 winters."

6. In Figure 1, The day 0 (March 09) are both in before-CAO and During-CAO periods. But Mar 10 is only in the During-CAO period. So, the definition of the period boundaries is not consistent. This may affect the analysis results (e.g., Figure 4 and so on).

**Response:** The day 0 is defined as the onset of a CAO event, which is the first day when regional mean cold airmass depth exceeds the threshold (166.8 hPa). Thus, for example, day +1 (−1) denotes 1 day after (before) the onset of the CAO event.

The definitions of different periods (before, during and after CAO) are intended to measure the reappearance of air pollution (see detailed definition in revised manuscript at Lines 112–118). We also correct the description of period during the CAO, which indeed starts from the onset day (day 0) and ends at the day when regional mean cold airmass depth falling below the threshold 166.8 hPa (Figure R3). To describe maximum and minimum values in the three periods more clearly and intuitively, we add some tags (AQI_b, AQI_d and AQI_a) in Figure R3. Results associated with the three periods are only shown in Figure 3 in the manuscript and will not affect the analysis results shown in Figure 4 and so on.

[Figure]

**Figure R3**. Time series of the regional averaged cold airmass depth (blue line) and AQI (orange line) in northern and eastern China (114°–122°E, 30°–40°N). The gray dashed line denotes the threshold value of the cold airmass depth.

In Figure 1/Figure R3, the maximum AQI in periods before (AQI_b) and after (AQI_a) CAO represent the original air pollution and reappeared air pollution, respectively. The

minimum AQI during CAO (AQI_d) represent a relatively clean state. Ideally, the period before CAO should not include day 0, which is partly coincide with the period during CAO. However, in some CAO events, the maximum AQI before CAO-related-reduction (AQI_b) occurs in day 0 (Figure R4). This is because the cold airmass at day 0 may only influence the northern part of northern and eastern China, although the regional mean cold airmass depth exceeds the threshold value. Our previous study has also shown that air pollutants will increase before the cold airmass arrives, and reach the maximum when the cold airmass just begins to invade (Liu et al., 2019). Therefore, to accurately measure the air quality before the reduction caused by CAO, we extend the period before CAO to day 0.

[Figure]

**Figure R4**. Evolutions of spatially averaged (northern and eastern China: 114°–122°E, 30°–40°N) cold airmass depth and AQI during a CAO event from 5 to 10 Dec 2017.

See revised text at Lines 116–121: "The onset of a CAO event, which is the first day when regional mean cold airmass depth exceeds the threshold (166.8 hPa), is described as the day 0. The period during the CAO starts from the onset day (day 0) and ends at the day when cold airmass depth falling below the threshold (day +2 in CAO event plotted in Figure 1). The period before CAO is defined as the two days before onset day to the onset day (days −2 to 0). The period after CAO, which is also called the decay phase, is defined as the three days after cold airmass depth falling below the threshold (days +2 to +4 in CAO event plotted in Figure 1)."

7. Figure 4, better to define the different color lines in the caption.

**Response:** Revised as your suggestion. See caption of Figure 4: "Blue lines and red lines denote the cold airmass depth and AQI, respectively".

We acknowledge your great help to improve the manuscript.

Thank you very much.

---

## Author Comment (AC2)

\*\*\*\*\*\*\*\*\*\*\*\*\*\*\*\*\*\*\*\*\*\*\*\*\*\*\*\*\*\*\*\*\*\*\*\*\*\*\*\*\*\*\*\*\*\*\*\*\*\*\*\*\*\*\*\*\*\*\*\*\*\*\*\*\*\*\*\*\*\*\*\*\*\*\*

In this file, **the text in black** shows the comments from reviewers and editor, while **the text in blue** is our replies.

\*\*\*\*\*\*\*\*\*\*\*\*\*\*\*\*\*\*\*\*\*\*\*\*\*\*\*\*\*\*\*\*\*\*\*\*\*\*\*\*\*\*\*\*\*\*\*\*\*\*\*\*\*\*\*\*\*\*\*\*\*\*\*\*\*\*\*\*\*\*\*\*\*\*\*

**SUMMARY:**

This work focused on the rapid increasing of air pollution after cold air outbreaks. The changes of AQI after CAO events is divided into two types of events: rapid change and slow change. By comparing and analyzing one example from each of the two types of events, the authors indicated that the depth, duration and coldness of cold air masses, as well as the stability of the vertical structure, modulated the changes in air pollution after the CAO events, and also made a comparative analysis of the role of thermal and dynamic processes. In addition, the two types of events corresponded to different atmospheric circulation systems, which may become previous signals. However, a lot of work mentioned about the drop of air pollution during the CAO and the subsequent rebound. The influence of the depth and duration of the cold air mass on the rebound of air pollution is also a classic conclusion in textbooks. Therefore, I think this research is not innovative enough. In addition, when analyzing the effects of cold air mass, the authors only conducted statistical analysis based on a single case, without the verification of numerical experiments, so its credibility and persuasion are not enough. A mandatory major revision is recommended.

**Response:** We would like to express our sincere thanks for your review efforts. We accepted all of your comments and revised the manuscript accordingly.

We agree that many previous studies have noted the reappearance of air pollution after CAO, while few studies have made statistics on the reappeared air pollution. So far, the quantitative relationship between air pollution reappearance and CAO properties and relevant physical mechanisms are still unclear. In this study, we show that the reappearance of air pollution after CAO is a common phenomenon. An isentropic analysis method (Iwasaki et al., 2014) is employed to quantitatively investigate the CAO of what features and structures could lead to the rapid reappearance of air pollution. Some large-scale patterns of CAO in its initial stage are also recognized as precursors for rapid reappearance of air pollution. We describe the novelty of this study at Lines 60–65.

To gain the credibility of our results, a series of numerical experiments were added in the manuscript as you suggested (See reply to MAJOR COMMENT #5).

The detailed response and revision are given below.

**MAJOR COMMENTS:**

1. Line 71-72: Air pollution in winter is mainly haze, but the data used in this study is AQI. I hope to know the change of PM2.5 during and after CAO and whether it is consistent with the conclusion of AQI

**Response:** We agree that air pollution in winter is mainly haze, during which the primary air pollutant is $PM_{2.5}$. Following your suggestion, we check the connection between AQI and $PM_{2.5}$ concentration. Figure R1 shows the daily mean AQI and $PM_{2.5}$ concentration in winters of 2014/2015–2021/2022 (2292 days). The correlation coefficient between them is as high as 0.96. We also investigate the evolutions of $PM_{2.5}$ concentration during the two types of CAO

events (Figure R2). The rapid and slow rebounds of PM₂.₅ concentration still could be observed in two kinds of CAOs. Therefore, the main conclusions of AQI is thought to be consistent with the results of PM₂.₅ concentration. We choose AQI as a comprehensive indicator in the manuscript as there are some cases that PM₂.₅ is not the primary air pollutant. We add above result in the manuscript at Lines 89–92: "It should be noted that the daily mean values of AQI and another well-known air pollution index of PM₂.₅ concentration (primary air pollutant of haze) has a high correlation coefficient of 0.96 in study period. The key results of this study are in a good agreement with results based on PM₂.₅ concentration (figure not shown)."

[Figure]

**Figure R1**. Relationship between daily mean AQI and PM$_{2.5}$ concentration averaged in North China.

[Figure]

**Figure R2**. Evolutions of spatial averaged (northern and eastern China: 114°–122°E, 30°–40°N) cold airmass depth and PM$_{2.5}$ concentration during CAOs. (a) and (b) are composited by the CAO_rapid events and CAO_slow events, respectively. Shading represents the 95% confidence interval of the composited mean value.

2. Lines 73-74: The data in this study is only used until 2018/2019. Why aren't the data for the last three years used? Especially in the winter of 2020/2021, many cold air events happened in North China, including record-breaking cold waves. The CAO events in the present study does not cover the recent years, lacking the timeliness of the facts.

**Response:** Following your suggestion, we expand the study period to 2014/2015–2021/2022. The number of CAO events increases to 52 including 33 rapid reappearance events and 19 slow reappearance events. The main results do not change after the study period is extended. We update Figures 2–6, 9, 12–13, Table 1 and relevant descriptions with new data (see revised manuscript).

3. Line 142-143: Only more than 50% of CAOs are found to show a worse AQI after reappearance. This ratio indicates that the change of AQI after CAO event is irregular, and the probability of its increase and decrease is basically the same. The phenomenon of air quality rapid rebounding after the CAO needs to be further confirmed.

**Response:** In this study, we find the reappearance of air pollution exists in the decaying period of all CAO events. To clarify the result of Figure 3, we would like to first explain the two indices ($IA$ and $RI$) defined in this study.

The deterioration rate of air quality during the reappeared air pollution is define by the change of AQI from the period during CAO to the period after CAO,

$$IA = AQI\_a - AQI\_d.$$

To describe the index more clearly and intuitively, we add some tags in Figure R3. $AQI\_a$ and $AQI\_b$ denote the maximum AQI in the periods after and before CAO, respectively $AQI\_d$ denote the minimum AQI in the period during CAO.

According to above definition, **the positive value of $IA$ indicates the reappearance of air pollution (e.g. Figure R3). Figure R4c shows that the $IA$ (x-axis) in all CAO events have a positive value, which confirms the reappearance of air pollution is a common phenomenon.** We add this result in the revised manuscript at Lines 167–168: "Figure 3c shows that the $IA$ (x-axis) in all of CAO events have a positive value, which means in AQI has experienced an increase after CAO. This result confirms the reappearance of air pollution is a common phenomenon."

[Figure]

**Figure R3**. Time series of the regional averaged cold airmass depth (blue line) and AQI (orange line)

in northern and eastern China (114°–122°E, 30°–40°N). The gray dashed line denotes the threshold value of the cold airmass depth.

Another variable *RI* is used to compare the air quality before and after CAO,

$$RI = AQI\_a/AQI\_b.$$

The value of *RI* larger (smaller) than 1 indicates the reappeared air pollution after CAO has a worse (better) air quality than the air pollution before CAO. Figure R4b shows more than half of CAOs will followed by a reappeared air pollution with worse air quality.

[Figure]

**Figure R4**. (a) Statistical distribution of daily change of AQI calculated by all of days during the 8 winters (brown bars) and days during the period after the CAOs (green bars). (b) Statistical distribution of the reappearance index of air pollution in 52 CAO events. (c) The increase in the AQI (x-axis) and the change in the AQI from the period before the CAO to the period after CAO (vertical arrows) in 52 CAO events. Black dashed line denotes the standard deviation of AQI. The arrowhead and tail represent AQI_a and AQI_b, respectively. The orange (blue) arrows denote that the AQI worsens (better) after CAO.

4. Lines 146-150: This part of the description is not very clear and hard to understand. The CAOs are divided into two groups: slow reappearance and rapid reappearance. In fact, not all AQI increased after CAO events. However, using "reappearance" to describe the two groups is misleading. In addition, the title of the abscissa in Figure 3c is "Increase of AQI after CAO", but actually there have decrease of AQI after some CAO events, so it is not appropriate to use "increase". In my understanding, the difference between the value of the arrowhead and tail should equal to the value of the abscissa, but obviously there are many points that do not correspond well. Please check the correctness of Figure 3c and add to the explanation in this part.

**Response:** In Figure R4c the position of each arrow on X-axis is *IA* ($AQI\_a - AQI\_d$). As explained in the response of MAJOR COMMENT #3, we found that in all of the CAO events AQI has experienced an increase. Here, we divide 52 events into two groups according to the value of *IA*. A rapid (slow) reappearance of air pollution is supposed to have an *IA* larger (smaller) than 34.3, which is the standard deviation of the AQI daily variation during the past 8 winters.

As for Y-axis, the difference between the values of arrowhead and tile is not equal to the value of abscissa. The position of arrowhead and tail on y-axis denote the *AQI\_b* and *AQI\_a*, respectively. Thus, the arrow on y-axis has a similar meaning with *RI*, indicating whether the air quality after CAO will get better or worse than that before CAO. The orange up arrow (blue down arrow) represents the reappeared air pollution has a heavier (lighter) degree than the air pollution before CAO. Combine the vertical direction of arrows with their positions on X-axis, we found the CAO_rapid (CAO_slow) events usually lead to a worse (better) air quality than that before CAO.

Following your suggestion, we add some detailed descriptions in the manuscript at Lines 173–176: "In Figure 3c, the arrow on y-axis has a similar meaning with *RI*, indicating whether the air quality after CAO will get better or worse than that before CAO. The position of arrowhead and tail on y-axis denote the *AQI\_b* and *AQI\_a*, respectively. The orange up arrow (blue down arrow) represents the reappeared air pollution has a heavier (lighter) degree than the air pollution before CAO."

5. Lines 189-190: When analyzing the effect of the depth and NFC of cold air mass on the rapid events and slow events, only one case is used for statistical analysis respectively, and the verification of numerical experiments is lacking. I am not sure whether the conclusion drawn through the individual case apply to most of the other rapid and slow events, so I am skeptical about the applicability and credibility of the effect of the depth and NFC of cold air mass. In addition, numerical experiments should be added to confirm the conclusion in Section 4.

**Response:** Following your suggestion, numerical experiments are added to verify our main results (See revised manuscript at Lines 305–340). First, the CAO_rapid event and CAO_slow event selected in Section 4 are simulated using WRF-Chem version 4.3. The simulated results are consistent with the observations. Then, a series of sensitive simulations are also conducted in which the connection between cold airmass properties and diffusion conditions of air pollutant is verified. These numerical experiments confirm that the CAO properties could determine the pollutant diffusion conditions in atmospheric boundary layer and further influence the rebounding speed of air pollution. The detailed analyses are listed below.

The domain of simulations is designed with a horizontal grid spacing of 10 km covering most part of East Asia (Figure R5). The FNL data is used as the initial and lateral boundary conditions to drive the meteorological simulation. The MEIC anthropogenic emission inventories are used in the chemical simulation. The main physical and chemical parameterization schemes include the WSM6 microphysics, the MYJ PBL scheme, the RRTM for longwave and shortwave radiation, RADM2-MADE/SORGAM for gas-phase chemical and aerosol schemes.

[Figure]

**Figure R5**. WRF model domain configuration (black rectangle). The colored shading presents the temperature disturbance in NHC_C experiment at model level 9 above ground. The blue rectangle represents the northern and eastern China.

Figure R6 shows the spatial averaged cold airmass depth, boundary layer height, vertical stability and surface PM$_{2.5}$ concentration during the CAO_rapid and CAO_slow events. Here, the emissions in both two experiments are set as the values in December 2016 to investigate the impacts of meteorological conditions. In experiment of CAO_rapid event, the air pollutant increases rapidly on days 0 and +1 under the condition of the relatively low boundary layer height and strong vertical stability. Such conditions of atmospheric boundary layer are not conducive to the diffusion of air pollutant and tend to induce rapid reappearance of air pollution (Zhang et al., 2014; Liu et al., 2017). In CAO_slow experiment, however, the PM$_{2.5}$ concentration keeps in a low level due to the high boundary layer height and weak vertical stability. In addition, the temporal evolutions of these variables are highly consistent with the observations shown in Figures 7–8 and 11 in the manuscript, suggesting both the rapid and slow reappearances of air pollution can be well captured by numerical model.

[Figure]

**Figure R6**. Evolutions of spatially averaged (northern and eastern China: 114°–122°E, 30°–40°N) cold airmass depth, atmospheric boundary layer height, vertical stability (1000–850 hPa) and surface PM$_{2.5}$ concentration during (a) CAO_rapid event from 14 to 17 Dec 2016 and (b) CAO_slow event from 11 to 14 Feb 2018.

To verify the connection between CAO properties and abovementioned boundary layer diffusion conditions as discussed in Sections 4.1 and 4.2, a control experiment (the CAO_rapid event in Figure 7) and additional sensitive simulations are also conducted. In sensitive experiments, temperature disturbances are artificially added in the initial field following Bai et al (2019). In NHC_C (NHC_W) experiment, the NHC of cold airmass is increased (decreased) by adding a cold (warm) bubble centered at a height of 0 km. The cold (warm) bubble had a latitudinal radius of 10 km, longitudinal radius of 5 km and a vertical radius of 2 km, with a minimum potential temperature perturbation of −8 (8) K. As shown in Figure R5, the temperature disturbance is minimized at the center and increased to 0 K following a cosine function over the horizontal and vertical radius. To increase (decrease) the cold airmass depth in DP_C (DP_W) experiment, the cold (warm) bubble added in the initial field moves to the height of 2 km. Note that the NHC may also change with cold airmass depth in DP_C and DP_W experiments.

Table R1 shows the simulation results averaged in study area on day 0, when air pollutant has the rapid increase rate as shown by Figure R6a. In NHC_C and NHC_W experiments, changes in NHC cannot cause an obvious variation in boundary layer height, but can lead to changes in vertical stability. In DP_C and DP_W experiments, despite the changes of NHC and vertical stability, we find that changes in cold airmass depth will result in an obvious change in boundary layer height. These sensitive experiments confirm the main results of Sections 4.1 and 4.2, that is, the properties of CAO could effectively impact the diffusion conditions in atmospheric boundary layer.

**Table R1:** Averaged cold airmass properties and atmospheric boundary layer conditions on day 0 from control and sensitive experiments. Upward (downward) arrows denote the value of sensitive experiment is greater (less) than the value in control experiment.

| Experiment | Cold airmass depth (hPa) | Atmospheric boundary layer height (m) | Negative heat content (− K hPa) | Vertical stability ($10^{-3}$ K m$^{-1}$) |
|---|---|---|---|---|
| Control | 165.6 | 597.2 | 1898.4 | 5.47 |
| NHC_C | 163.9 | 579.9 | 1994.5 ↑ | 6.79 ↑ |
| NHC_W | 167.6 | 661.0 | 1745.9 ↓ | 4.15 ↓ |
| DP_C | 189.4 ↑ | 727.4 ↑ | 2447.9 | 5.98 |
| DP_W | 118.8 ↓ | 519.9 ↓ | 1238.4 | 4.74 |

**References**

Bai L, Meng Z, Huang Y, Zhang S, Niu S and Su T 2019 Convection initiation resulting from the interaction between a quasi-stationary dryline and intersecting gust fronts: A case study *J. Geophys. Res. Atmos.* **124** 2379–2396

Liu Q, Sheng L, Cao Z, Diao Y, Wang W and Zhou Y 2017 Dual effects of the winter monsoon on haze-fog variations in eastern China *J. Geophys. Res. Atmos.* **122** 5857–5869

Zhang R, Li Q and Zhang R 2014 Meteorological conditions for the persistent severe fog and haze event over eastern China in January 2013 *Sci. China Earth Sci.* **57** 26–35

**MINOR COMMENTS:**

1. Lines 30-33: The use of tenses is confusing, alternating between the general present tense and the past tense. Suggest to check the whole text and unify tenses.

**Response:** Following your suggestion, we check the whole text and unify tenses.

2. Lines 28-30: What does the air pollution mainly refer to in the study? Is it haze pollution? Many articles about haze pollution are cited in the formation of air pollution in the following paragraph. Be more specific about the characteristics of air pollution.

**Response:** The air pollution in winter over North China is mainly haze pollution. Following your suggestion, we specify the type of air pollution in the introduction at Lines 29–30: "North China, experiences the most pollution, which mainly is haze pollution, resulting in attention from researchers."

3. Lines 40-41: impacting the variations in air pollution [on the synoptic time-scale].

**Response:** Revised as your suggestion. See Lines 42–43: "As emissions and topography do not vary much from day to day, meteorological conditions play an important role in impacting variations in air pollution on the synoptic time-scale."

4. Line 50: increased as much as 2.8 times than what?

**Response:** Following your suggestion, we revise the sentence at Lines 52–53: "During the reappearance of air pollution, the regional mean pollutant concentrations increased as much as 2.8 times than concentrations reduced by CAO."

5. Line 100: defined as the two days before onset day [to] the onset day

**Response:** Revised as your suggestion. See Lines 118–119: "The period before CAO is defined as the two days before onset day to the onset day (days −2 to 0)."

6. Lines 100-101: Does "the three following the period during the CAO" refer to day 1 to 4? It does not match the annotation in Figure 1.

**Response:** "The three days following the period during the CAO" refer to the three days after the end date of period during the CAO, when regional mean cold airmass depth falling below the threshold (166.8 hPa). It should be noted that the period after CAO varies with events, since the end date of period during the CAO is different in the selected events. Thus, in the CAO event plotted in Figure 1 (Figure R3), "the three days" is days +2 to +4. According to your comment, we add an explanation at Lines 119–122: "The period after CAO, which is also called the decay phase, is defined as the three days after cold airmass depth falling below the threshold (days +2 to +4 in CAO event plotted in Figure 1). The period after CAO varies with events, since the end date of period during the CAO is different in the selected events."

7. Lines 142-144: This view should be presented with caution. It is probably because the

selection time of the period before CAO avoids the high value AOI, which may have been affected by cold air.

**Response:** Following your suggestion, we revised this sentence as: "Based on the definitions of periods before and after CAO in our study, more than 50% of the CAOs are found to show a worse AQI after reappearance. In some extreme events, the AQI after CAO could be twice as high as the AQI before CAO."

8. Figure 5 and 6: Add the meaning of the black box in the caption.

**Response:** Following your suggestion, we add the meaning of black box in the caption of Figure 5 at Line 574: "The black boxes denote the northern and eastern China"

9. Figure 10: what does the dots refer to?

**Response:** The dots in Figure 10 refer to the area with vertical gradient of the potential temperature $> 6 \times 10^{-3} \ K \ m^{-1}$. See caption of Figure 10 at Lines 587: "vertical gradient of the potential temperature (contour, unit: $10^{-3} \ K \ m^{-1}$, dotted for $> 6 \times 10^{-3} \ K \ m^{-1}$)"

We acknowledge your great help to improve the manuscript.

Thank you very much.

---

## Author Response (AR2)

\*\*\*\*\*\*\*\*\*\*\*\*\*\*\*\*\*\*\*\*\*\*\*\*\*\*\*\*\*\*\*\*\*\*\*\*\*\*\*\*\*\*\*\*\*\*\*\*\*\*\*\*\*\*\*\*\*\*\*\*\*\*\*\*\*\*\*\*\*\*\*\*\*\*\*

In this file, **the text in black** shows the comments from reviewers and editor, while **the text in blue** is our replies.

\*\*\*\*\*\*\*\*\*\*\*\*\*\*\*\*\*\*\*\*\*\*\*\*\*\*\*\*\*\*\*\*\*\*\*\*\*\*\*\*\*\*\*\*\*\*\*\*\*\*\*\*\*\*\*\*\*\*\*\*\*\*\*\*\*\*\*\*\*\*\*\*\*\*\*

**Editor decision from Dr. Paul Zieger:**

**SUMMARY:**

Thank you for your revised version. The reviewers are generally satisfied with your changes and only one short comment by reviewer #3 concerning the re-analysis data should be considered further before we can finally accept your manuscript.

Dear Editor,

We would like to express our sincere thanks to you and reviewers for the kind evaluation. We accept the comments and revised the manuscript accordingly.

We agree with the reviewer that model grid reanalysis data has a similar vertical resolution with sounding data. The model grid reanalysis data may be a good substitute of sounding data in some circumstances. We also explain the necessity of using sounding data to validate the reliability of reanalysis data in this study.

Thank you again for your consideration.

Sincerely Yours,

Guixing Chen, Ph.D.

On behalf of the authors of ACP-2022-9

September 19, 2022
* * *
In this file, **the text in black** shows the comments from reviewers and editor, while **the text in blue** is our replies.
* * *
**Reviewer #3**

**COMMENTS:**

The authors well addressed my previous comments. However, I still have some questions related to the usage of JRA-55 reanalysis data. I would like to recommend this manuscript be considered for publication after these questions have been addressed.

Based on the information provided in the replies, the authors use JRA-55 1.25-degree latitude/longitude grid data (LL125). However, JRA-55 also has 6-hourly reanalysis data on model grid data (TL139, https://jra.kishou.go.jp/JRA-55/document/JRA-55_handbook_TL319_en.pdf). TL139 has 60 vertical levels, which have 11 levels under 850hPa. Model variables include geopotential height, temperature, u/v-component of wind, specific humidity, etc. Although the vertical levels are not in the isobaric surfaces as LL125, the authors can still do the interpolation under the hydrostatic equilibrium assumption. On the other hand, back to my previous comment 1.2, since TL139 has 11 levels under 850hPa, this is similar to the sounding data, which has 8-14 levels below 850hPa.

I also suggest that the authors list the data source for all variables used in this manuscript in a Table. e.g., summarize which variables are from JRA-55 and which variables are from the local station.

**Response:** Thank you for your advice. We agree that the model grid data of JRA-55 reanalysis (TL319L60) has a similar vertical resolution with sounding data. Our results also suggest the JRA-55 reanalysis data and sounding data have good consistency in describing vertical structure of atmospheric boundary layer. Thus, the model grid reanalysis data is thought to be a good substitute of sounding data in our future work.

Here. we would like to explain why we use sounding data in this study. In section 4.1, the sounding data is employed to verify whether reanalysis data could well capture the structure of atmospheric boundary layer, and whether the depth (DP) of cold airmass could represent the mixing layer height (MLH). Since the calculation of DP is already based on JRA-55 reanalysis data, the MLH used for comparison is better to be calculated by observing data such as sounding data rather than JRA-55 reanalysis data. On the other hand, the sounding data provides a direct detection of atmospheric vertical profiles, which could be used to validate the reliability of model-simulated reanalysis data. Due to above concerns, the analysis of sounding data and its comparison with reanalysis data is remained in this study.

Accordingly, we rephrase the relevant text in section 2.1: "The sounding data obtained from the University of Wyoming provides a direct detection of atmospheric vertical profile, which would help to explain the changes of AQI observation. Four sounding stations were selected in NEC: Beijing (39.8°N, 116.5°E), Zhangqiu (36.7°N, 117.6°E), Nanjing (31.9°N, 118.9°E) and Baoshan (31.4°N, 121.5°E). Observation times were 00 and 12 UTC (08 and 20 LT). The vertical resolution of the sounding data is comparable to that of JRA-55 model grid

reanalysis data. For example, sounding data at Beijing station has 60–70 levels in total and 8–14 levels below 850 hPa during a CAO event of 14–17 Dec 2016."

Following your suggestion, we also add a table (Table R1) showing the data source of variables used in this study in section 2.1. See revised text at Line 98: "The data sources of all variables used in this study are listed in Table 1."

**Table R1:** Data source of variables used in this study. Here, $u$: zonal winds, $v$: meridional winds, $T$: air temperature, $\Phi$: geopotential height, $p_s$: surface pressure.

| Data source | Variables |
|---|---|
| JRA-55 reanalysis data (Japan Meteorological Agency) | $u, v, T, \Phi$ (1000~100 hPa) and $p_s$ |
| Air quality monitoring data (Ministry of Ecology and Environment of the People Republic of China) | AQI |
| Radio sounding data (University of Wyoming) | $u, v, T, \Phi$ (surface to 100 hPa) |

We acknowledge your great help to improve the manuscript.

Thank you very much.